# Exploring the characteristics of FY-4A/AGRI visible reflectance using the CMA-MESO forecasts and its implications to data assimilation

Yongbo Zhou[1,2], Yubao Liu[1,2], Wei Han[3,4], Yuefei Zeng[5], Haofei Sun[6], Peilong Yu[5,7,8], Lijian Zhu[9]

[1]School of Atmospheric Physics, Nanjing University of Information Science & Technology, Nanjing, China

[2]Precision Regional Earth Modeling and Information Center (PREMIC), Nanjing University of Information Science & Technology, Nanjing, China

[3]CMA Earth System Modeling and Prediction Centre (CEMC), Beijing, China

[4]State Key Laboratory of Severe Weather (LaSW), Beijing, China

[5]Key Laboratory of Meteorological Disaster of Ministry of Education, Collaborative Innovation Center on Forecast and

Evaluation of Meteorological Disasters (CIC-FEMD), Nanjing University of Information Science & Technology, Nanjing, China

[6]Institute of Atmospheric Physics, Chinese Academy of Science, Beijing, China

[7]College of Meteorology and Oceanography, National University of Defense Technology, Changsha, China

[8]Key Laboratory of high Impact Weather (special), China Meteorological Administration, Changsha, China

[9]Shanghai Typhoon Institute, China Meteorological Administration, Shanghai, China

*Correspondence to*: Yongbo Zhou (yongbo.zhou@nuist.edu.cn)

**Abstract.** The Advanced Geostationary Radiation Imager (AGRI) onboard the Fengyun (FY)-4A geostationary satellite provides high spatiotemporal resolution visible reflectance data since 12 March 2018. Data assimilation experiments under the framework of observing system simulation experiments have shown great potential of these data to improve the forecasting

skills of numerical weather prediction (NWP) models. To assimilate the AGRI visible reflectance in real-world cases, it is important to evaluate the quality and to quantify the observation errors of these data. In this study, the FY-4A/AGRI channel 2 (0.55 μm - 0.75 μm) reflectance data ($O$) were compared with the equivalents ($B$) derived from the short-term forecasts of the China Meteorological Administration Mesoscale (CMA-MESO) Model using the Radiative Transfer for the Television infrared observation satellite Operational Vertical Sounder (TOVS) (RTTOV, v12.3). It is shown that the $O - B$ biases could

be used to reveal the abrupt change related to the measurement calibration processes. In general, the $O - B$ departure was positively biased in most cases. Potential causes include the deficiencies of the NWP model, the forward-operator errors, and the unresolved aerosol processes, etc. The relative biases of $O - B$ computed from cloud-free and cloudy pixels were used to correct the systematic biases for the corresponding scenarios over land and sea surfaces separately. In general, the method effectively reduced the $O - B$ biases. Moreover, the bias-correction method based on an ensemble forecast is more robust

than a deterministic forecast due to the advantages of the former in dealing with uncertainties in cloud simulations. The findings demonstrate that analysing the $O - B$ biases has a potential to monitor the performance of FY-4A/AGRI visible instrument and to correct the systematic biases in the observations, which will facilitate the assimilation of these data in conventional data assimilation applications.

## 1. Introduction

The Advanced Geostationary Radiation Imager (AGRI) is one of the main payloads onboard the Fengyun (FY)-4A, the first of the new-generation Chinese geostationary meteorological satellites launched on 11 December 2016 (Yang et al., 2017). FY-4A/AGRI contains seven shortwave channels and seven infrared channels. The radiance observations ranging from visible to infrared channels have been widely used to retrieve cloud optical thickness (Chen et al., 2020), total precipitable water (Liu et al., 2022), and Aerosol Optical Depth (AOD) (Ding et al., 2022). In addition, the FY-4A infrared radiance data were

assimilated into Numerical Weather Prediction (NWP) models, and positive impacts on the forecasts of Typhoon cases (Zhang et al., 2022) and Heavy rainfall events (Xu et al., 2023) were reported. The FY-4A/AGRI visible radiance data were also assimilated under an Observation System Simulation Experiment (OSSE) framework, and the results revealed positive impacts on cloud variables and some slightly positive impacts on non-cloud variables in the vicinity of cloudy regions (Zhou et al., 2022; Zhou et al., 2023).

The AGRI, with minor improvements by including an extra infrared channel, was also equipped to the FY-4B, which is the second of the Chinese new-generation geostationary meteorological satellites launched on 3 June 2021. FY-4A and FY-4B were initially located at 104.7 °E and 133.0 °E, respectively. The two satellites cover a large part of the East Asian and Western Pacific, providing rich visible and infrared radiance data that are highly valuable for data assimilation applications. From February 1 to 5 March 2024, FY-4B drifted from 133.0 °E to 104.7 °E to replace the FY-4A and started its operational

observations from 00:00 UTC on 5 March 2024. Since the visible instruments onboard the two satellites share similar characteristics, the general findings of one satellite could be extended to another one.

Data assimilation of the FY-4A/AGRI radiance data in real cases demands accurate description of the Probability density Distribution Function (PDF) of the observation errors. Conventional data assimilation methods assume that the observations are unbiased and the PDF of the observation errors conforms to a Gaussian function (Geer and Bauer, 2011; Bonavita et al.,

2016; Li et al., 2022). The observation errors influence the data assimilation results by tuning the weights given to each observation. Several techniques were deployed to characterize the systematic biases of satellite observations and an inter-comparison method between the satellite observations (*O*) and the equivalents (*B*) derived from the forecasts of NWP models using forward operators received general popularity, especially for the satellite infrared and microwave channels (Auligné et al., 2007; Zou et al., 2016; Lu et al., 2020; Noh et al., 2023). Unlike variational and Ensemble Kalman Filter (EnKF)

(and its variants) methods, it is unnecessary for the particle filter method to make a Gaussian distribution assumption for the PDF of observation errors. Nevertheless, one topic of this study is to explore the bias correction of the visible reflectance under the framework of variational and EnKF data assimilation methods since they are the mainstream data assimilation methods in the NWP centres worldwide.

The inter-comparison method was also applied to satellite visible channels (Geiss et al., 2021; Lopez and Matricardi, 2022;

Lopez et al., 2022) to better understand the observation errors and representativeness errors and to provide guidance for the improvements of NWP models and forward operators. Most of the studies performed the radiative transfer simulations based on a software package termed the Radiative Transfer for the Television infrared observation satellite Operational Vertical Sounder (TOVS) (RTTOV) (Saunders et al., 2018). To save computational cost, a method for fast satellite image synthesis (MFASIS) was developed based on a lookup table (LUT) computed with one-dimensional (1D) solver in rotated Cartesian coordinates to account for some three-dimensional (3D) radiative effects (Scheck et al., 2016; Scheck et al., 2018). To better simulate the tangent linear and adjoint models, a neural network-based forward operator was also developed based on 1D radiative transfer simulations (Scheck et al., 2021). Intercomparison of satellite visible reflectance and the equivalents derived from NWP models and MFASIS indicated generally good agreement, and the Bidirectional Reflectance Distribution Function (BRDF) of land surface derived from a monthly mean atlas generated reasonable results in cloud-free conditions (Lopez and Matricardi, 2022). However, neglecting aerosol contributions in the radiative transfer simulations would lead to systematic biases both in cloudy and cloud-free conditions (Geiss et al., 2021). Data assimilation of satellite visible reflectance data based on the MFASIS suggested positive impacts in real-world cases (Scheck et al., 2020). Since March 2023, satellite visible reflectance data have been operationally assimilated in German Weather Service by using the MFASIS forward operator. Existing studies imply the promising expectation that RTTOV could generate reliable visible images if the NWP models were well tuned and the model configurations were optimized.

One assumption of the inter-comparison method is that the spatiotemporal characteristics of different error contributions differ so that the $O - B$ analysis can be used to identify different error sources. NWP models face challenges to generate representative atmosphere state variables due to their inherent limitations such as the deficiencies of microphysical schemes, the uncertainties of the initial conditions (ICs) and lateral boundary conditions (LBCs), the unresolved sub-grid processes, etc (Janjić et al., 2017). The errors in NWP models could be alleviated by assimilating synergic observations with improved data assimilation methods, or by ensemble forecasts which involve several microphysics combinations or different ICs and LBCs (Li et al., 2015), etc. In addition, forward operators inevitably suffer from errors due to the uncertainties of cloud optical properties (Zhou et al., 2018), aerosol-cloud interactions (Geiss et al., 2021), and the BRDF of sun-glint areas over sea surface, etc. To save computational cost, 3D radiative processes were usually simplified into 1D processes, which is another source of forward-operator errors. The main factors which contribute to the simulation errors of the reflectance equivalents should be properly assessed to increase the robustness of the inter-comparison results.

In this study, the FY-4A/AGRI channel 2 (0.55 μm - 0.75 μm) reflectance data were compared with the equivalents derived from the forecasts of the China Meteorological Administration Mesoscale (CMA-MESO) model using RTTOV (v12.3). The main purpose of this study is to address the following two questions. First, is analysing the $O - B$ departure an effective way to monitor the performance of FY-4A/AGRI visible instrument? Second, what are the characteristics of the $O - B$ departure and how to correct the systematic biases in O in order to assimilate satellite reflectance data in real-world

cases? In view of the two questions, the remaining part of this manuscript is organized as follows. Data and method were introduced in Section 2. Results were presented in Section 3. Uncertainties due to forward operators and unresolved aerosol processes were discussed in Section 4. Implications to correct the systematic biases in $O$ for data assimilation were explored in Section 5. Conclusions were summarized in Section 6.

## 2. Data and method

### 2.1 Simulated CMA-MESO visible reflectance

Simulated visible reflectance was generated from the forecasts of the CMA-MESO model, which is the operational mesoscale model in CMA. The domain coverage of the CMA-MESO model is shown by Fig. 1, which includes 2501 × 1671 horizontal grids with a grid spacing of 0.03° and 50 vertical layers with a model top of 10 hPa. The main physical configurations of the CMA-MESO model include the Single-Moment 6-class microphysical scheme (Hong and Lim, 2006), the Meso-SAS (Simplified Arakawa-Schubert) shallow-convective cumulus parameterization option (Zhang et al., 2017), the Yonsei University (YSU) planetary boundary layer scheme (Hong and Lim, 2006; Hu et al., 2013), the Unified Noah land surface scheme (Tewari et al., 2004), and the Rapid Radiative Transfer Model for Global Climate Models (RRTMG) longwave and shortwave radiation schemes (Iacono et al., 2008). The model configurations generate non-cloud variables (water vapour mixing ratio, temperature, etc.) and cloud variables including the mixing ratio of five cloud hydrometeors (cloud droplet, rain, ice, snow, and graupel) and cloud cover.

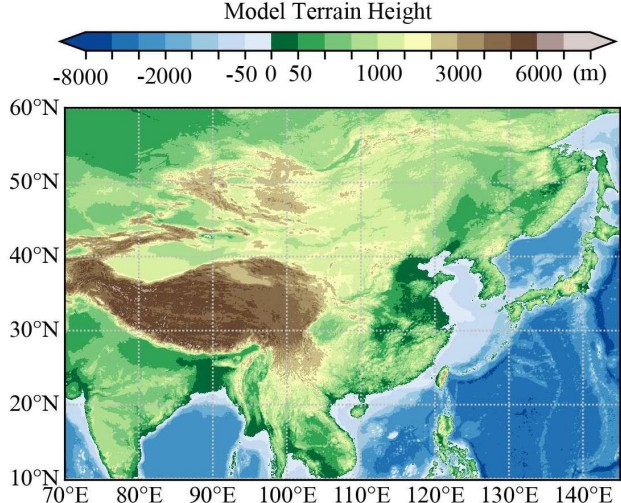

**Figure 1: The domain coverage of the CMA-MESO model, which includes 2501×1671 horizontal grids with a horizontal grid spacing of 0.03°.**

Previous studies suggested that the parameterization for unresolved sub-grid clouds was critical to the simulated reflectance (Scheck et al., 2018; Geiss et al., 2021). In this study, the sub-grid clouds were approximated by the meso-SAS shallow-convective cumulus parameterization. The tendency equations of the grid-box mean moist static energy, water vapour mixing ratio, and vertical velocity were related to the transfer equations of related variables at sub-grid scale. The mixing ratio

of cloud hydrometeors at sub-grid scale was generated by convective condensation with interactions to gird-scale processes considered. The spatial coverage of the sub-grid clouds within a grid box was depicted by cloud cover, which was diagnosed from the grid-scale humidity following Xu and Randall (1996). The cloud cover derived from the CMA-MESO forecast was included in the RTTOV input to account for the sub-grid contributions and the radiative transfer was solved by using the maximum random overlap method.

The 6-h forecasts of CMA-MESO at 06:00 UTC in September were used to generate synthetic visible images for comparison with corresponding observations. To generalize the results, the intercomparison was also performed in March, June, and December and the results were provided in the supplementary material. The following discussions would refer to the results in September unless otherwise specified. In order to extend the bias correction method to EnKF-like methods, an ensemble forecast at 06:00 UTC was constructed by including seven deterministic forecasts with different forecasting lead times of 3 h, 6 h, 9 h, 12 h, 15h, 18 h, and 21 h, respectively. The 6-h and 18-h forecasts, which were initialized at 00:00 and 12:00 UTC on the previous day, respectively, got their ICs and LBCs from the large-scale background field provided by the CMA-Global Forecasting System (GFS). Other ensemble members got the ICs and LBCs from the analysis fields which were generated by assimilating the cloud motion wind retrieved from the observations of FY-2G (one of the Chinese first-generation geostationary satellites) and Himawari-8 (the first Japanese next-generation geostationary satellite), the Global Navigation Satellite System (GNSS) radio occultation (RO) data, the FY-4A/AGRI clear-sky infrared radiances, etc (Shen et al., 2020). The synergic observations were assimilated by a 3D variational (3DVar) data assimilation system.

It is noted that the ensemble forecast here could not represent a real ensemble forecast in any operational ensemble DA systems. On one hand, the number of ensemble members is too small to fully represent the uncertainties of atmosphere states. On the other hand, a more commonly used way to generate an ensemble forecast is to add perturbations to the ICs and LBCs or to combine several forecasts with different combination of microphysical schemes (Li et al., 2015). The simplified ensemble forecast in this study was used mainly because none of a well-tuned ensemble forecast is currently available for the selected area. In a real ensemble DA system, real ensemble members would be adopted for the bias correction. Synthetic visible images derived from the ensemble forecast should be accompanied with increased cloud cover since clouds are not exactly overlapped for different ensemble members, i.e., the displacement errors. As a result, the number of matched pixels which are cloudy both for the observations and simulations would be increased, which benefited the bias correction in cloudy regions (see Section 5 for more details).

The forecasts of CMA-MESO were processed into the format of the RTTOV input files to facilitate the radiative transfer simulations. The sun-viewing geometries (i.e., solar zenith angles, satellite zenith angles, and relative azimuth angles between the sun and satellite sensor) were derived from the FY-4A synchronous observation geometry (GEO) data gridded at 4 km × 4 km resolution, which were interpolated to the CMA-MESO grids by a bilinear interpolation. The layer-to-space transmittance was computed by the v9 predictors on 54 levels (Matricardi, 2008). The BRDF was drawn from monthly mean land surface

atlases (Vidot and Borbás, 2014; Vidot et al., 2018) or calculated by the Joint North Sea Wave Project (JONSWAP) model for the sea surface (Hasselmann et al., 1973). The radiative transfer processes were solved by the Discrete Ordinate Method (DOM) with 16 streams. The liquid and ice cloud optical properties in RTTOV were parameterized by the "Deff" scheme (Mayer and Kylling, 2005) and the Baran et al. (2014) scheme, respectively. Since the state variables of the CMA-MESO model does not include the effective radius of liquid water clouds ($Re_{liq}$) and ice clouds ($Re_{ice}$), $Re_{liq}$ was explicitly calculated following Thompson et al. (2004) and Yao et al. (2018). $Re_{ice}$ was not calculated explicitly since the ice scheme developed by Baran et al. (2014) does not have an dependence on $Re_{ice}$.

## 2.2 FY-4A/AGRI visible reflectance and cloud mask

To generate spatially collocated observations and simulations, the FY-4A/AGRI full-disk channel 2 reflectance data gridded at 1 km×1 km resolution were horizontally averaged to the CMA-MESO locations. The horizontal averaging was performed by the following two procedures. Firstly, centring at a given CMA-MESO grid point and finding all the pixels (matched pixels hereafter) in the FY-4A/AGRI visible image within ± 0.015° both in the zonal and meridional directions. Secondly, averaging the reflectances of all these matched pixels to generate a reflectance that is spatially matched to the CMA-MESO grid. Repeating the two steps for all CMA-MESO grid points generated an observed image gridded at 0.03°×0.03°. The full-disk scanning cycle of AGRI is 15 minutes and the scanning usually starts at 00:00 UTC. In addition, the CMA-MESO forecasts were produced at hourly intervals. Therefore, the maximum allowable time differences between the FY-4A observations and CMA-MESO forecasts are within 15 minutes to ensure the temporal match. In addition, the 4 km × 4 km FY-4A cloud mask (CLM) product were used to provide a first-step estimate of cloud or clear sky. Since the CLM product contains discrete values, the 4 km × 4 km CLM data were matched to the CMA-MESO location by the least-distance matching. After applying the above-mentioned processes to the FY-4A level 1 observations, the FY-4A visible reflectance data and CLM data were spatiotemporally matched to the CMA-MESO simulations. Fig. 2 shows an example of the FY-4A/AGRI observations matched to the CMA-MESO grids, including the visible reflectance of channel 2, cloud mask, solar zenith angle, solar azimuth angle, satellite zenith angle, and satellite azimuth angle.

## 2.3 The multi-source observed precipitation products gridded at 1 km resolution

Since the representativeness of $B$ was collaboratively determined by the CMA-MESO forecasts and RTTOV-DOM simulations, it is important to evaluate the performance of CMA-MESO to better understand the $O - B$ statistics. The forecasts of CMA-MESO were evaluated by the multi-source observed precipitation products, which provide one-hour accumulated precipitation over the whole Chinese mainland with a horizontal resolution of 0.01° ($\approx$ 1 km) (Pan et al., 2018). The products were produced by the National Meteorological Information Center (NMIC) using the hourly precipitation data of

nearly 40,000 automatic weather stations in China, the Quantity Precipitation Estimate (QPE) retrieved from radar (Liu et al., 2010), and the precipitation product retrieved by the National Oceanic and Atmospheric Administration (NOAA) Climate Prediction Center Morphing Technique (CMORPH) (Joyce et al., 2004). To develop the merged precipitation product, the hourly precipitation observations from the automatic weather stations were interpolated to 0.01°×0.01° grid points by the optimal interpolation method. The 0.01°×0.01° data were then merged with the precipitation products from the QPE and CMORPH based on the Bayesian Model Averaging method. To ensure the spatial collocation between the observations and simulations located in the CMA-MESO grids, the merged precipitation product was horizontally averaged to the CMA-MESO locations by the same methods introduced in Section 2.2.

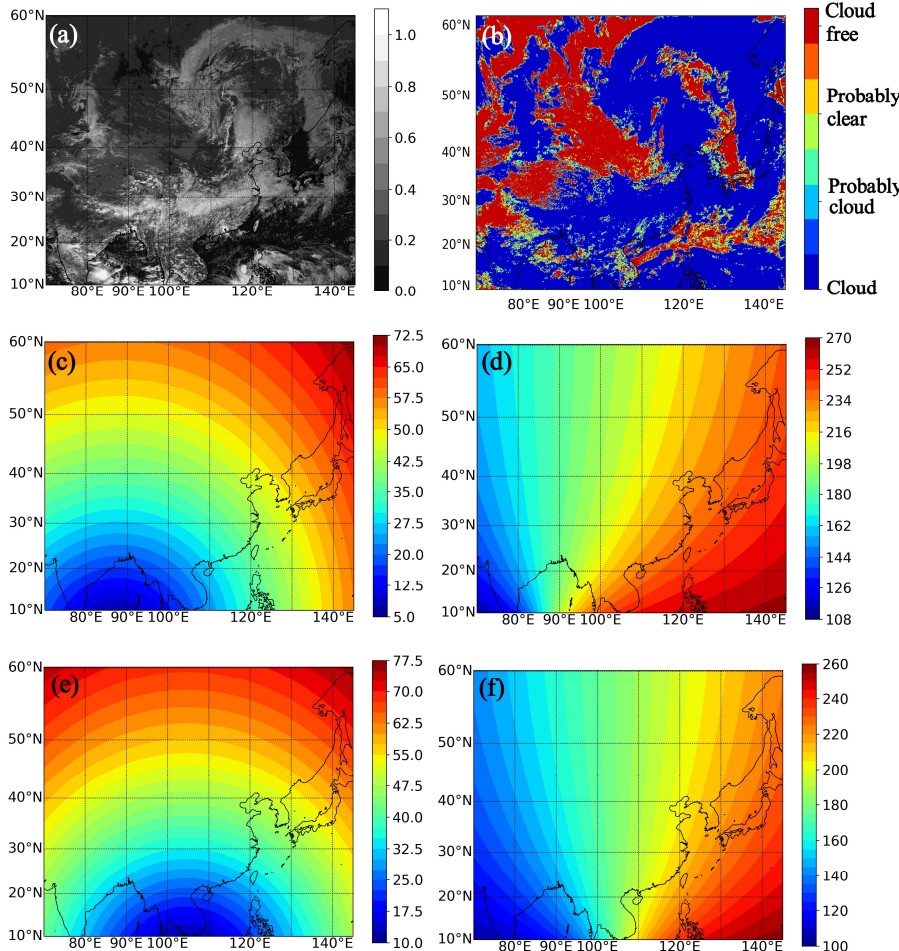

**Figure 2: FY-4A/AGRI observations at 06:00 UTC on 15 September 2020, which were matched to the CMA-MESO grids. (a) Reflectance at 0.65 µm; (b) Cloud mask derived from the FY-4A CLM product; (c) Solar zenith angle; (d) Solar azimuth angle; (e) Satellite zenith angle; (f) Satellite azimuth angle.**

## 3. Results

### 3.1 Evaluation of CMA-MESO forecasts

A comparison of the one-month mean 1-h accumulated precipitation at 06:00 UTC for the observations and simulations was shown by Fig. 3. In general, good consistency between the simulations and observations was revealed, except that the

precipitation areas were overestimated by the CMA-MESO forecasts in Chinese mainland. Since the 6-h forecast was cold-started, the overestimation of precipitation was probably caused by the biases in the LBCs and ICs downscaled from the CMA-Global Forecasting System (GFS) fields at 00:00 UTC or by the deficiencies of the CMA-MESO model itself. To illustrate this problem, the PDFs of one-month Brightness Temperature (BT) for the FY-4A/AGRI channel 13 (10.30 μm – 11.30 μm) observations and simulations were analysed following the guidance of Geiss et al. (2021). The BT simulations were done with RTTOV-DOM with the same configurations introduced in Section 2.1. The results were shown in Fig. 4. For BT simulations, the PDF was underestimated at the high-BT end. In contrast, it was overestimated at the low-BT end. Since channel 13 is an infrared window channel, BT in cloudy areas is directly related to cloud top height. Therefore, the PDF analysis implies that high-level clouds were underestimated by CMA-MESO whereas low-level clouds were overestimated. A potential explanation is that the shallow cumulus parameterization in CMA-MESO model could lead to an underestimation of the convective weather system compared with real cases (Wan et al., 2015).

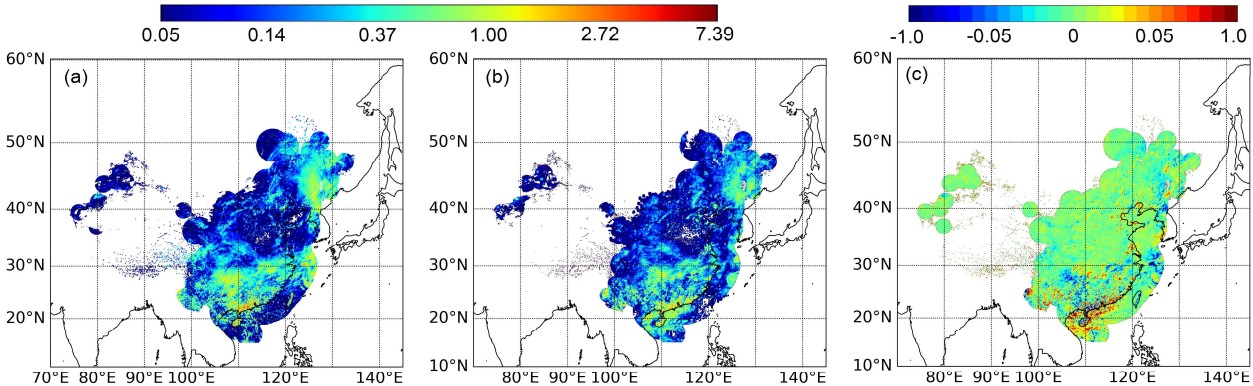

**Figure 3: The one-month mean 1-h accumulated precipitation (unit: mm) at 06:00 UTC in September for the (a) simulations from the 6-h forecasts of CMA-MESO model, and (b) observations from the multi-source observed precipitation products, and (c) the one-month mean observations minus simulations.**

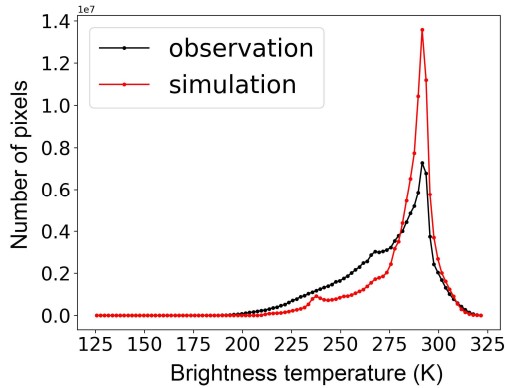

**Figure 4: The probability density distribution function of the one-month brightness temperature at 06:00 UTC in September for the FY-4A/AGRI observations at channel 13 and the corresponding simulations.**

The deficiencies of CMA-MESO in forecasting high-level clouds do not necessarily mean that the synthetic reflectance is under- or overestimated. The Top-Of-Atmosphere (TOA) reflectance is mainly determined by CWP and the

effective radius of cloud particles. In contrast, precipitation depends not only on the two parameters, but also on the cloud vertical structures, cloud phases, etc. In addition, the variation of the reflectance would become saturated when CWP reaches to a threshold value (e.g., Fig. 4(c) in Zhou et al., 2023), whereas precipitation is likely to be positively related to CWP (Wang et al., 2024). Nevertheless, the time series of the domain-averaged precipitation for CMA-MESO forecasts agreed well with the observations, except that the CMA-MESO forecasts were overestimated (Fig. S1 in the supplementary material). For the deterministic forecasts with a forecasting lead time of 3 h, 9 h, 12 h, 15 h, and 21 h, the CMA-MESO model was warm-started, with cloud initial fields created by a cloud analysis technique. The cloud analysis technique tended to introduce false-alarm cloud hydrometeors in the initial fields. As a result, the short-term forecasts of CMA-MESO tended to produce false-alarm precipitation and the precipitation tended to be overestimated (Zhu et al., 2017).

**3.2 Spatial distribution of *O-B* biases**

The spatial distribution of the one-month $O - B$ biases in September 2020 was shown in Fig. 5. The results for March, June, and December were shown in Fig. S2 in the supplementary material. The results indicated that positive biases were especially apparent over the Southern foothills of the Himalayas, the Sichuan basin, and the Yunnan-Kweichow Plateau. On one hand, some areas of the Qinghai-Tibet Plateau were covered with snow. Reflectance simulated in these areas should be less accurate compared with other places since the BRDF atlas is questionable in snow-covered areas (Ji et al., 2022). On the other hand, the performance of the CMA-MESO model was reduced over complex terrain areas. To illustrate this, the observed and synthetic images and their corresponding PDFs for two typical cases were shown by Fig. 6. Based on subjective evaluation of grayscale image tones, the model missed some clouds over the southern part of the Qinghai-Tibet Plateau and Sichuan Basin (Fig. 6(a1-a2)). In addition, some of the orographic clouds over the southern slope of the Himalayas were missed (Fig. 6(b1-b2)). In the central areas of a cyclone system, the simulations generated some gaps which were actually filled with clouds in the observations (Fig. 6(a1-a2)). The comma-shaped clouds along the southern China were also underestimated by the simulations (Fig. 6(b1-b2)).

The PDF analysis of the two cases revealed that the number of pixels for the reflectance smaller than around 0.1 was larger for the simulations than the observations (Fig.6 (a3) and Fig. 6 (b3)). The pixels with reflectance smaller than 0.1 mainly represent cloud-free pixels. The low-reflectance end of the PDF was shifted toward the left, mainly because cloud cover was underestimated by the simulations, as was confirmed by the observed and synthetic images. For cloud-free pixels, the presence of aerosols tends to increase the TOA reflectance due to the extra photons backscattered to the satellite by aerosols (Geiss et al., 2021), which should be another explanation to the left-tilted PDF for the simulations when reflectance is less than 0.1. On the contrary, the number of pixels was underestimated by the CMA-MESO forecasts for a medium reflectance ranging from 0.1 ~ 0.4 or 0.1 ~ 0.6 for the two cases, respectively. There are many potential causes to such an underestimation. For example, the CMA-MESO model could underestimate the convective clouds compared with real cases (Wan et al., 2015). Since typical

cyclone systems were presented for the two cases, the CMA-MESO's deficiency in simulating strong convection should be an important cause to the underestimation of PDF in the medium reflectance range. For optically thick cloud (reflectance > 0.6), the PDF of the simulations agrees well that of the observations. The variation of reflectance with CWP becomes saturated when the CWP reaches to a threshold value. As a result, the impacts of the NWP model errors would be mitigated for thick clouds.

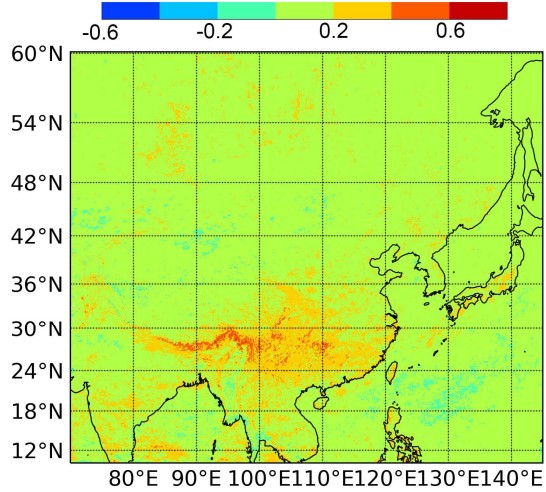

**Figure 5: Spatial distribution of the *O-B* biases in September for FY-4A channel 2.**

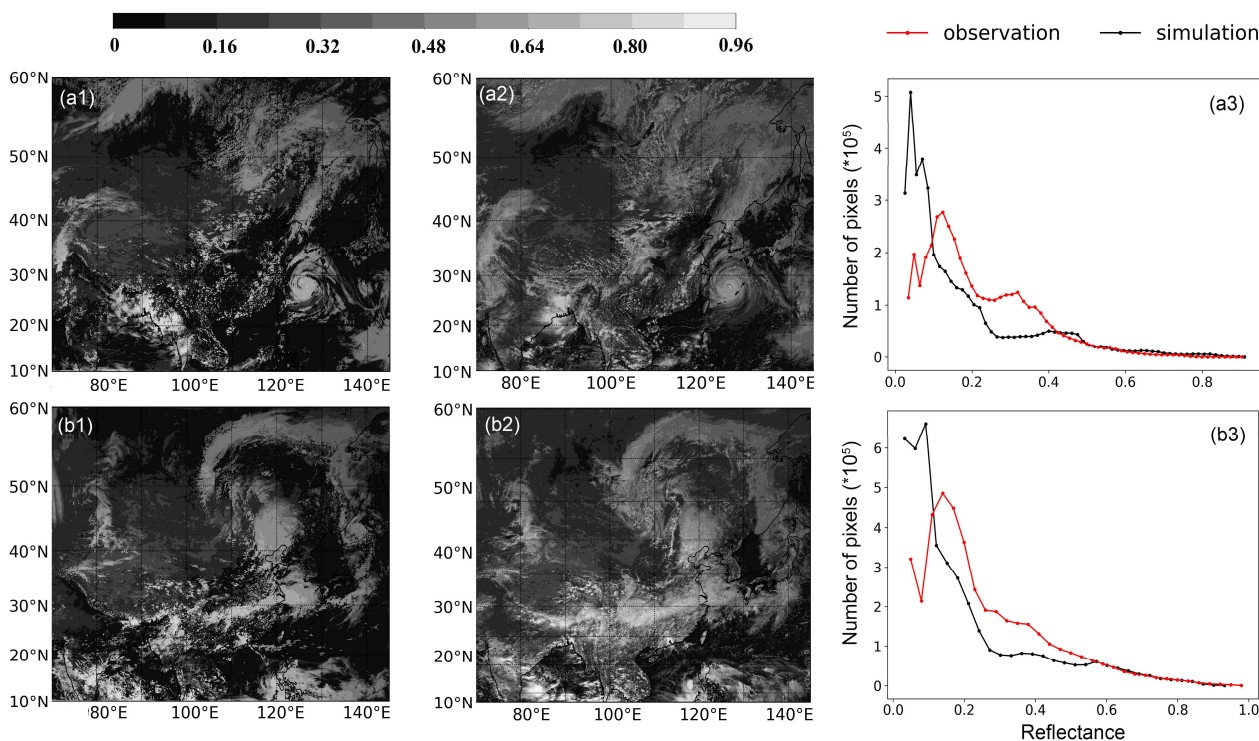

**Figure 6: Synthetic (the first column) and observed (the second column) visible images and the corresponding probability density distribution functions (the third column) for two selected cases. The first panel (a1-a3) is the results for the case at 06:00 UTC on 1 September 2020. The second panel (b1-b3) is the results for the case at 06:00 UTC on 15 September 2020.**

## 3.3 Temporal variation of *O-B* biases

In order to better understand different error contributions to the $O - B$ biases, it necessary to exclude some pixels in the selected domain where the representativeness errors are especially large.

First, a threshold test of terrain height was applied to exclude the Qinghai-Tibet Plateau areas by Equation (1),

$$ter \leq 4.0$$

(1)

where $ter$ denotes the terrain height (unit: km) in the CMA-MESO domain. The threshold value 4.0 is the mean terrain height of the Qinghai-Tibet Plateau.

Second, the snow-covered areas were screened out by applying a threshold test of the surface albedo. The surface albedo of snow in the visible spectral band varies with the physical properties of snow. With the increase of the average radius of ice grains, the surface albedo is decreased (Gardner and Sharp, 2010). In addition, the surface albedo of dirty snow, which includes absorbing particles, and old snow, which includes some melting water, is smaller than that of pure snow (Xu and Tian, 2000; Gardner and Sharp, 2010). In general, the lower limit of the surface albedo for snow-covered surfaces in China is suggested to be 0.2 (e.g., Fig. 3 of Xu and Tian, 2000). Therefore, the threshold test was performed by Equation (2),

$$\omega \leq 0.2/3.14$$

(2)

where $0.2/3.14$ denotes the BRDF for a Lambertian radiator.

Third, the highly reflective areas over sea surface, i.e, the sun-glint areas, were excluded to reduce the representativeness errors in these areas. Although sophisticated algorithm for locating the sun-glint areas were developed (e.g., Li et al., 2009), a simple threshold test could identify most of the sun-glint areas in this study,

$$B_{clr}^{sea} > 0.1$$

(3)

where $B_{clr}^{sea}$ denotes the clear-sky reflectance simulated by RTTOV-DOM. In this case, the inputs to RTTOV were derived from CMA-MESO forecasts, except that the mixing ratio of cloud hydrometeors was set to zero.

Since both the observation errors in $O$ and the representativeness errors in $B$ were cloud-dependent, the $O - B$ analysis was performed for the cloudy and cloud-free pixels separately. Unlike Geiss et al. (2021) where a threshold value of 0.2 was applied to determine whether a pixel is cloudy or cloud-free, the cloud mask in this study was determined by comparing the simulated and observed reflectance with the reflectance simulated by ignoring cloud impacts. For synthetic visible images, a pixel was designated to be cloudy if Equation (4) was satisfied. Otherwise, the pixel would be designated to be cloud-free.

$$B_{clm}^{sfc} > B_{clr}^{sfc}$$

(4)

where $B_{clm}^{sfc}$ denotes the simulated reflectance. The subscript $clm$ denotes cloud mask, which is either cloud-free ($clr$) or cloudy ($cld$). The superscript $sfc$ denotes the surface type that is either land or sea.

The aerosol contributions were neglected by the simulations since the CMA-MESO forecasts do not provide aerosol information explicitly, whereas the observed reflectance inevitably includes aerosol contributions. For the observed image, a pixel was designated to be cloudy if the observed reflectance $O_{clm}^{sfc}$ satisfied Equation (5),

$$O_{clm}^{sfc} > B_{clr}^{sfc} + r_{aer}^{75} \qquad (5)$$

where $r_{aer}^{75}$ denotes the aerosol contributions to the reflectance of cloud-free pixels, which was set to the upper quartile of

$O_{clr}^{sfc} - B_{clr}^{sfc}$ for the preliminarily estimated cloud-free pixels. In addition, $O_{clr}^{sfc}$ denotes the observed reflectance for the preliminarily estimated cloud-free pixels. The second-step estimate of cloud-free pixels was determined Equation (6),

$$O_{clm}^{sfc} < B_{clr}^{sfc} + r_{aer}^{25} \qquad (6)$$

where $r_{aer}^{25}$ denotes the aerosol contributions to the cloud-free reflectance. Similarly, $r_{aer}^{25}$ was set to the lower quartile of

$O_{clr}^{sfc} - B_{clr}^{sfc}$ for the preliminarily estimated cloud-free pixels. The two-step estimate of cloud mask in observed images was

performed to maintain equivalent criterion of the cloud mask for synthetic images. It is noted that the first-step estimate of cloud mask should be different from that diagnosed from Equation (4). For example, the CLM product was generated by including extra infrared observations (Wang et al., 2019) that are much more sensitive to optically thin cloud, which may appear to be transparent in the visible band. Nevertheless, the quartile estimation should mitigate the impacts. On one hand, thin clouds which are transparent in the visible channel whereas are opaque in the infrared channels should contribute

insignificant part to $O_{clm}^{sfc}$. On the other hand, the quartile estimation in Equations (4) and (5) discarded 25% samples in estimating the lower and upper quartiles of $O_{clr}^{sfc} - B_{clr}^{sfc}$.

        After excluding the cases with noticeable representativeness errors, the one-month temporal variation of the $O$ - $B$ biases in September was shown by Fig. 7. The results for March, June, and December was shown in Fig. S3 - Fig. S5 in the supplementary material. The results indicate that the $O$ - $B$ biases in cloudy regions are especially large compared those in

cloud-free regions. Therefore, the $O$ - $B$ biases originated mainly from observation errors or representativeness errors in cloudy regions. The representativeness errors were determined by the NWP model errors and RTM errors, which are particularly evident in cloudy conditions due to the deficiencies of NWP models in modelling clouds (Lopez and Matricardi, 2022) or the uncertainties in cloud optical properties (Geiss et al., 2021). In addition, there are substantial differences of the $O$ - $B$ biases between land and sea surfaces. The differences between $O$ and $B$ were closely related to the performance of

CMA-MESO model over land and sea surfaces due to the parameterization schemes, the data effectively used by the 3DVAR system or the cloud analysis technique, etc. Nevertheless, the $O$ - $B$ biases were mainly determined by the results over land due to the predominant pixels therein.

        Although the comprehensive contributing factors make the $O$ - $B$ statistics rather complicated, some of the error sources could be revealed from the $O$ - $B$ analysis. For example, an abrupt change of the bias from September 8[th] to 9[th] was revealed

in Fig. 7. The abrupt change was caused by the measurement calibration processes. In fact, the calibration correction coefficient of FY-4A/AGRI channel 2 was updated by the National Satellite Meteorological Center (NSMC) of CMA at 02:00 UTC on 9 September 2020 (http://www.nsmc.org.cn/nsmc/cn/news/103609.html) (remember that both the observations and

simulations were deployed at 06:00 UTC). Since the $O$ - $B$ biases were positively related to the observed reflectance (not shown for simplicity) which is proportional to the calibration coefficient, the abrupt change was amplified for cloudy pixels compared with the cloud-free pixels. The finding answers the first question that analysing the $O$ - $B$ departure is an effective method to monitor the performance of the FY-4A visible instruments. After the update of the calibration correction coefficient, the absolute values of the biases were reduced for cloud-free pixels (Fig. 7(b)). Since the radiative transfer simulations are more reliable for cloud-free pixels than for cloudy pixels (Lopez and Matricardi, 2022), the results confirmed the effectiveness of the calibration processes. In contrast, the absolute values of the biases were increased for cloudy pixels mainly due to uncertainties associated with cloud state variables and the cloud optical properties which will be partly discussed in Section 4.1.

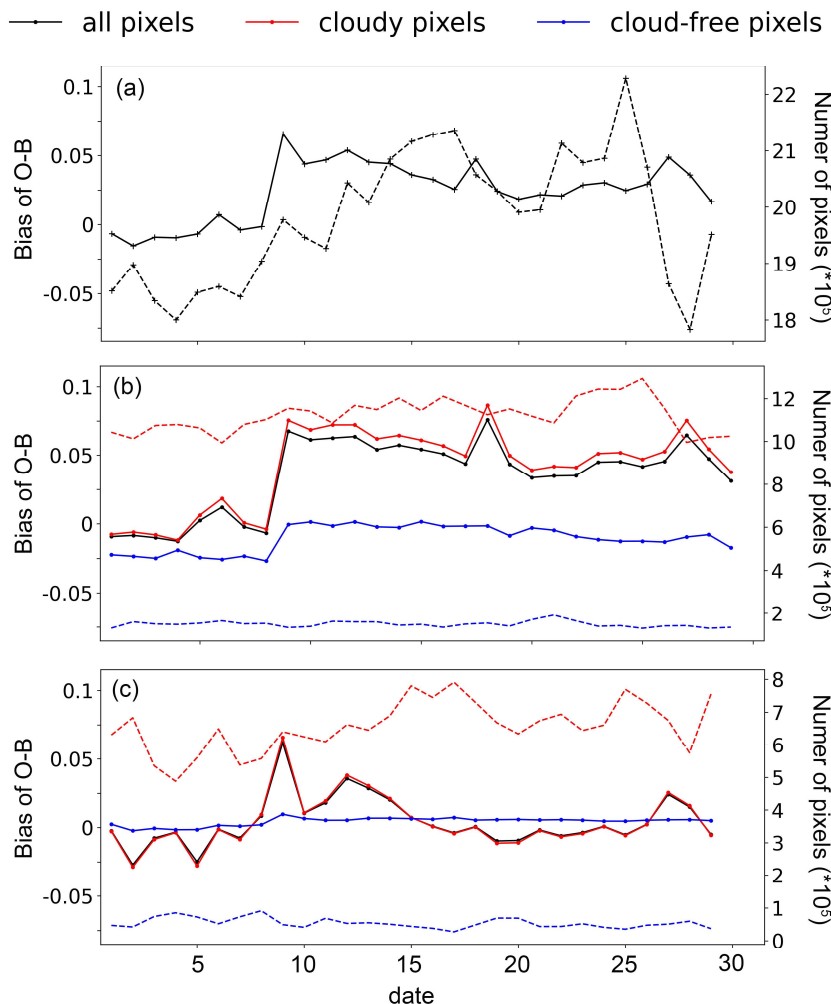

**Figure 7: The time series of the *O-B* biases for all (cloudy + cloud-free), cloudy, and cloud-free pixels in September 2020. The results are shown for (a) all underlying surfaces (including land and sea), (b) land surface, and (c) sea surface. The dashed lines denote the number of pixels for different cloud mask and underlying surfaces.**

It is noted that another abrupt change was also revealed on June 21[th] (Fig. S4(b)). The abrupt change was caused by the annular solar eclipse on 06:00 UTC 21 June 2020, when the incoming solar radiance was sheltered by the moon over the west parts of the CMA-MESO domain. The annular solar eclipse caused an abrupt decrease of the photons received by the AGRI visible channel. As a result, the visible image was darkened. The darkened visible image was also revealed by the National

Aeronautics and Space Administration (NASA) worldview project (https://worldview.earthdata.nasa.gov/). However, the annular solar eclipse was not considered when performing the radiative transfer simulations by RTTOV-DOM. Instead, the incoming solar irradiance was set to a constant, which caused an abrupt decrease of the $O$ - $B$ biases.

    In addition to the error sources mentioned above, the $O$ - $B$ biases were collaboratively determined by many other factors. For example, $O$ contains the impacts of aerosols which could not be reflected by $B$ since the CMA-MESO cannot

resolve aerosols processes. The spatiotemporal variations of aerosols are evident (Liu et al., 2019) and should have non-negligible impacts on the $O$ - $B$ biases. Besides, the $O$ - $B$ biases were influenced by the performance of the forward operator, which is subject to many factors such as the accurate description of cloud optical properties for the liquid water clouds and ice clouds. Detailed discussions on the influences of all the main contributing factors will be given in Section 4.

## 4. Uncertainties due to forward-operator errors and unresolved aerosols

**4.1 Forward-operator errors**

Another main contributing factor to the errors in $B$ is the forward operator, i.e., the RTTOV-DOM in this study. For example, the pre-assumed cloud Particle Size Distribution (PSD) inherent in the cloud schemes in RTTOV is inconsistent with that of NWP models, not to mention the representativeness of the pre-assumed PSD in real cases. These problems will inevitably introduce errors to the synthetic visible images (Yuan et al., 2022).

Currently, there are many alternative parameterization schemes of optical properties for liquid water clouds and ice clouds in RTTOV. For example, RTTOV-DOM includes the ice cloud optical properties schemes developed by Baum et al. (2011) and Baran et al. (2014), respectively (Baum and Baran schemes hereinafter for simplicity). The Baum scheme calculated cloud optical properties, including the scattering phase function, single scattering albedo, and extinction coefficient, based on the mixing ratio of ice hydrometeors and $Re_{ice}$. In comparison, the Baran scheme did not explicitly rely on $Re_{ice}$. Instead, the

optical properties were parameterized by the mixing ratio of ice hydrometeors and the temperature. The Baum and Baran schemes were declared to be applicable to the ice water content ranging from 4.98 $\times 10^{-5}$ to 0.1831 $gm^{-3}$ and 6.0$\times 10^{-6}\sim$ 1.969466 $gm^{-3}$, respectively (Hocking et al., 2016). The forecasts of CMA-MESO showed that the ice water content exceeds the valid range of the Baum schemes in some cases. Therefore, RTTOV-DOM was configured with the Baran scheme in this study, but this does not necessarily mean that the Baran scheme outperforms Baum scheme. Sophisticated evaluation will be

needed to address the performance of each scheme in real cases.

    The impacts of ice cloud schemes on the simulated reflectance were illustrated by a sensitivity study performed by RTTOV-DOM configured with the Baum and Baran schemes based on the 6-h forecasts of CMA-MESO at 06:00 UTC on September 1[st] and 15[th], 2020. To facilitate the radiative transfer simulations for the Baum scheme, $Re_{ice}$ was estimated from the CMA-MESO forecasts by Equation (7) (Hong et al., 2004; Yao et al., 2018),

$$Re_{ice} = \min(11.9 \times 0.75 \times 0.163 \times M_i^{1/2}, 500 \times 10^{-6})$$ (7)

where $M_i$ denotes the ice crystal mass, which was calculated by Equation (8),

$$M_i = \frac{\rho_a q_i}{N_i}$$ (8)

where $\rho_a$ denotes the density of air. $q_i$ denotes the mixing ratio of ice crystals. $N_i$ denotes the concentration of ice

crystals which was approximated by Equation (9),

$$N_i = \min(\max(5.38 \times 10^7 (\rho_a \times max(q_i, 10^{-15}))^{0.75}, 10^3), 10^6)$$ (9)

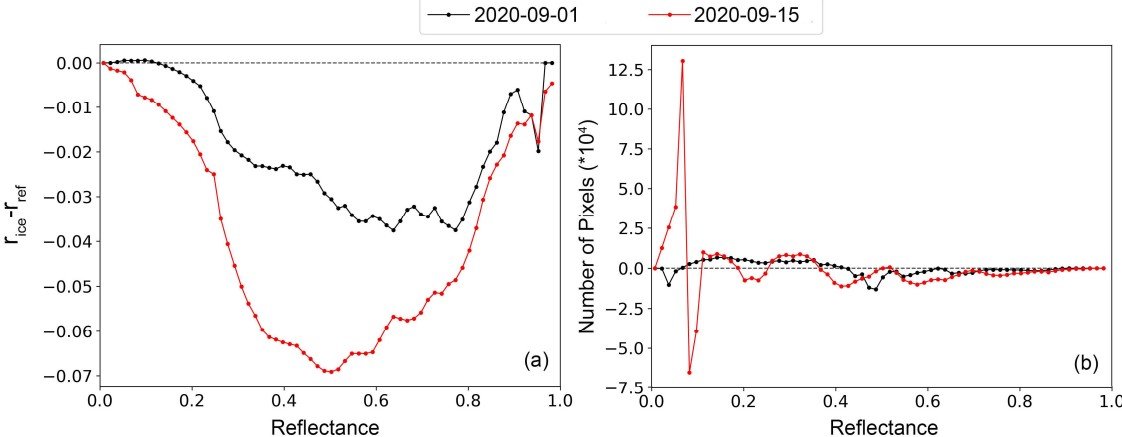

**Figure 8: (a) The biases of reflectance simulated by RTTOV-DOM configured with the ice scheme of Baum et al. (2011) ($r_{ice}$) and the reference run ($r_{ref}$) which is configured with the ice scheme of Baran et al. (2015). (b) Differences between the reflectance PDFs obtained from the simulations the ice scheme of Baum et al. (2011) and the reference run (the former minus the latter).**

The results indicated that the reflectance simulated by Baum scheme was underestimated compared with that simulated

by the Baran scheme (the reference run hereafter) (Fig. 8(a)). The differences between the FY-4A visible reflectance PDFs

obtained from the simulations based on the Baum scheme and the reference run indicated that the impacts were especially

apparent for optically thin clouds (reflectance < 0.2) (Fig. 8(b)) and extended to optically thick clouds. In the high-reflectance

end, the PDF was underestimated by the reference run compared with the simulations based on the Baum scheme. The results

are different from Geiss et al. (2021) which suggested that changing the ice scheme from the general habit mixture (GHM)

developed by Baum et al. (2014) to a solid-column scheme based on ice optical properties of Yang et al. (2005) only affected

the high-reflectance end of the PDF. We did not conduct an inter-comparison study of ice cloud schemes between the solid

columns and GHM. But Baum et al. (2014) compared the ice cloud optical thickness retrieved based on the GHM and solid

columns and indicated good consistency between the two ice models due to their similar asymmetry parameters. The ice cloud

optical properties were determined by ice habits, PSDs, the mixing ratio of each habit, etc. Substantial differences exist when

building the bulk scattering properties of Baum and Baran schemes. For example, the Baum scheme was developed based on

nine basic ice habits whereas the Baran scheme involves only six ice habits. In addition, the PDFs and the mixing ratio of each

habit are different between the two ice schemes, which could lead to non-negligible differences between the two ice models.

Therefore, the distinct differences between the Baran and Baum schemes should be the main cause to the larger differences

than Geiss et al. (2021) between the reference run and experiment run. The results imply the uncertainties in the cloud optical

properties of RTTOV-DOM.

As is mentioned above, 3D radiative effects also contribute to the forward-operator errors and they could be alleviated by

increasing the model grid spacing (Várnai and Marshak, 2001; Zinner et al., 2006) or simply by horizontally averaging of the

pixels (Kostka et al., 2014). However, small-scale properties could not be properly resolved with large grid spacing or could be

cancelled out for the observations averaged over n × n pixels (n denotes the number of pixels involved). In view of this, the

horizontal averaging was not performed to the 0.03°×0.03° forecasts or the observations.

## 4.2 Unresolved aerosol processes

The aerosol processes cannot be properly resolved by the CMA-MESO model. However, aerosols have significant impacts on

the observed reflectance, which is the theoretical basis for the remote sensing of AOD by satellite observations. To evaluate the

impacts of aerosols on the TOA reflectance, a sensitivity study was performed by RTTOV-DOM with varying aerosol optical

properties based on the 6-h forecast of CMA-MESO at 06:00 UTC on 15 September 2020. The aerosols were assumed to

decrease with height exponentially with a scale height of 2.0 km. The optical properties of aerosols were configured with those

of the dust aerosol of Cloud-Aerosol Lidar and Infrared Pathfinder Satellite Observation (CALIPSO) (Omar et al., 2009). The

optical properties of the dust aerosol at the central wavelength of FY-4A/AGRI channel 2 (0.65 μm) were calculated by a

logarithmic interpolation of the optical properties at 0.532 μm and 1.064 μm provided by Zhou et al. (2017). The logarithmic

interpolation was also used to supply the AOD out of the reference wavelengths in the SBDART radiative transfer model

(Ricchiazzi et al., 1998). Since the radiative transfer simulations were rather time-consuming when aerosol contributions were

considered, only 10,000 atmospheric columns within the CMA-MESO domain were randomly chosen for the sensitivity study.

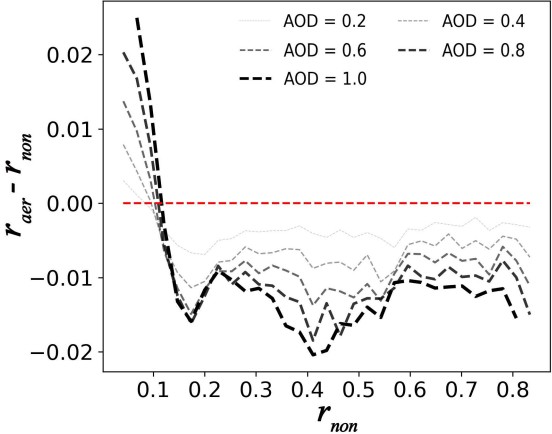

**Figure 9: The impacts of dust aerosols on the TOA reflectance. $r_{non}$ denotes the reflectance simulated by RTTOV-DOM based on cloud profiles derived from the 6-h forecast at 06:00 UTC on 15 September 2020. $r_{aer}$ denotes the reflectance based on cloud profiles and dust aerosols.**

The impacts of dust aerosols on the TOA reflectance are demonstrated by Fig. 9. The results indicate that the impacts are

highly dependent on AOD and CWP (As a general approximation, CWP is positively related to the TOA reflectance). Under

cloud-free conditions, the presence of dust aerosols tends to increase the TOA reflectance due to the fact that dust aerosols scatter some photons to the satellite sensors. With the increase of CWP, the impact of dust aerosols tends to generate negative bias on the TOA reflectance. A potential explanation is that dust aerosols absorb some photons from the incoming path to clouds and from the outgoing path to satellite. The two-fold impacts of aerosols were also reported by Geiss et al. (2021). The aerosol contributions were also related to aerosol types and aerosol vertical distribution structures. Since the satellite visible

reflectance is especially sensitive to cloud properties, data assimilation of visible reflectance data has been designed to adjust cloud variables (e.g., the mixing ratio of a cloud hydrometers, cloud cover, and effective radius of cloud particles, etc.) rather than the aerosol properties (Scheck et al., 2020; Zhou et al., 2022; Zhou et al., 2023). In this case, aerosol contributions could be deemed as noises to the observations. The results in Fig. 9 indicate that aerosols introduced systematic biases to the TOA reflectance, and the influences are distinctly different for cloudy and cloud-free pixels. Therefore, it is possible that the

aerosol-induced noises in reflectance observations could be corrected and that the bias correction should be tackled for the cloudy and cloud-free pixels separately.

## 5. Implications to bias correction for data assimilation

For the data assimilation of satellite infrared and microwave data, the equivalents derived from the first-guess forecasts of NWP models have been used as a reference to correct the systematic biases in observations. Some well-designed predictors

such as the average cloud impact (Harnisch et al., 2016) or the NWP model state variables (Noh et al., 2023) were regressed to the $O - B$ biases and the systematic biases were corrected based on the regression.

    Compared with the infrared and microwave radiance observations, the visible reflectance is much more sensitive to cloud variables, regardless of the type of cloud hydrometeors or the vertical location of clouds. In contrast, the infrared radiance data are only sensitive to cloud-top properties due to strong absorption effects (Li et al., 2022). The microwave radiance data are

insensitive to small cloud hydrometeors and were usually used to constrain large particles such as rain drops (Wang et al., 2021). In addition, the visible reflectance is less sensitive to temperature or humidity compared with the infrared and microwave radiances. Since the NWP model errors are particularly evident in cloudy conditions (Mathiesen and Kliessl, 2011) and the predictor-based bias correction is largely determined by the equivalents derived from NWP forecasts, the robustness of the predictor-based bias correction method should be reduced when applied to the visible spectral bands. In view of the

analyses above, the systematic biases in $O - B$ was simply corrected by the first-order approximation method promoted by Harnisch et al (2016), i.e., the mean difference of $O - B$, denoted by $\overline{O - B}$ where the bar denotes the domain averaging. To avoid generating reflectance beyond the $0 \sim 1$ range during the bias correction, the first-order approximation of the $O - B$ bias was depicted by $\overline{O - B} / \overline{O}$ rather than $\overline{O - B}$. Therefore, the bias-corrected reflectance $O'$ is calculated by Equation (10),

$$O' = O(1 + \gamma_{clm}^{sfc}) \tag{10}$$

where $\gamma_{clm}^{sfc}$ denotes the bias correction coefficient. In addition to the denotations of $clm$ in Equation (4), the $clm$ in Equation (10) also represents uncertain ($uct$) scenarios which were designated by the FY-4A CLM product. Therefore, $\gamma_{clm}^{sfc}$ represents one of the six bias correction coefficients including $\gamma_{clr}^{land}$, $\gamma_{cld}^{land}$, $\gamma_{uct}^{land}$, $\gamma_{clr}^{sea}$, $\gamma_{cld}^{sea}$, and $\gamma_{uct}^{sea}$. For a deterministic forecast, the bias correction coefficients were calculated by Equations (11)-(16),


$$\gamma_{clr}^{land} \leftarrow \frac{\sum_{k=1}^{N_{clr}^{land}} [O_{clr}^{land}(k) - B_{clr}^{land}(k)]}{\sum_{k=1}^{N_{clr}^{land}} O_{clr}^{land}(k)} \tag{11}$$

$$\gamma_{cld}^{land} \leftarrow \frac{\sum_{k=1}^{N_{cld}^{land}} [O_{cld}^{land}(k) - B_{cld}^{land}(k)]}{\sum_{k=1}^{N_{cld}^{land}} O_{cld}^{land}(k)} \tag{12}$$

$$\gamma_{uct}^{land} = (\gamma_{clr}^{land} + \gamma_{cld}^{land}) / 2 \tag{13}$$

$$\gamma_{clr}^{sea} \leftarrow \frac{\sum_{k=1}^{N_{clr}^{sea}} [O_{clr}^{sea}(k) - B_{clr}^{sea}(k)]}{\sum_{k=1}^{N_{clr}^{sea}} O_{clr}^{sea}(k)}$$

(14)


$$\gamma_{cld}^{sea} \leftarrow \frac{\sum_{k=1}^{N_{cld}^{sea}} [O_{cld}^{sea}(k) - B_{cld}^{sea}(k)]}{\sum_{k=1}^{N_{cld}^{sea}} O_{cld}^{sea}(k)} \tag{15}$$

$$\gamma_{uct}^{sea} = (\gamma_{clr}^{sea} + \gamma_{cld}^{sea}) / 2 \tag{16}$$

where $N_{clm}^{sfc}$ denotes the number of matched pixels between $O$ and $B$ for a specific surface type and cloud mask. $k$ is the index for an arbitrary pixel.

For the EnKF-like methods, the observation increments were calculated using the ensemble mean in the observation space (e.g. Equation (6) of Anderson, 2010). To maintain consistency with the ensemble-based DA methods, the bias correction method should be performed based on the ensemble mean of the first-guess reflectance, denoted by $\overline{B_{clm}^{sfc}}$, which was generated by Equation (17),

$$\overline{B_{clm}^{sfc}} = \frac{1}{N_{ens}} \sum_{l=1}^{N_{ens}} B_{clm}^{sfc}(l) \tag{17}$$

where $N_{ens}$ denotes the number of ensemble members. $l$ is the index for an arbitrary ensemble member.

To correct the biases in $O$ according to Equation (10), $\gamma_{clm}^{sfc}$ was calculated for the cloud-free or cloudy pixels for land and sea surfaces separately. The implication of Equation (10) is that the systematic biases estimated from the matched pixels which are cloud-free (or cloudy) for both $O$ and $B$ were extended to the cloud-free (or cloudy) pixels only for $O$. Apparently, the cloud-free (or cloudy) pixels both for $O$ and $B$ are only a subset of those only for $O$. Therefore, the performance of the bias correction is determined by the representative of the subset of the matched cloud-free (or cloudy) pixels to the corresponding cloud-free (or cloudy) pixels only in the observed images.

The bias correction based on deterministic and ensemble forecasts was tested by two selected cases on September 15$^{st}$ and 17$^{th}$, 2020. For the ensemble forecast, cloud mask was determined by Equation (4) except that $B_{clm}^{sfc}$ and $B_{clr}^{sfc}$ were replaced

by the ensemble mean values. For the bias correction based on a deterministic forecast, the biases of $O$ - $B$ were reduced in most cases, but increased biases were also revealed on September 17[th] for cloudy regions over sea (Table 2). In contrast, the bias reduction was especially effective when $B$ was derived from ensemble forecasts (Table 1 and Table 2). For the bias correction based on an ensemble forecast, the ensemble averaging could decrease the reflectance for a pixel classified to be cloudy for a deterministic forecast due to the displacement errors (i.e., some of the ensemble members were cloud-free while others are cloudy). Therefore, the bias correction coefficient estimated by the ensemble forecasts is larger than that estimated by a deterministic forecast. Nevertheless, there should have some advantages for the ensemble forecast. For example, if clouds occur for all the ensemble members, the uncertainty of the ensemble mean should be smaller than that of a single ensemble member. Since the number of the matched cloudy pixels was larger for the ensemble forecast than the deterministic forecast, $\gamma_{cld}^{sfc}$ derived from ensemble forecasts should represent cloudy bias characteristics better than a deterministic forecast, which explains why the biases were increased in some cases based on deterministic forecasts.

For cloud-free pixels, the ensemble averaging will increase the reflectance compared with the reflectance for a deterministic forecast. As a result, $r_{aer}^{25}$ estimated from an ensemble forecast was increased compared with that estimated from a deterministic forecast. Consequently, the ensemble averaging tends to increase the reflectance for a pixel classified to be cloud-free for a deterministic forecast, leading to the increased bias correction coefficients for the ensemble forecast. In addition, the number of the matched cloud-free pixels was smaller for the ensemble forecast than the deterministic forecast. As a result, $\gamma_{clr}^{sfc}$ derived from a deterministic forecast should represent cloud-free bias characteristics better than an ensemble forecast, which explains why the bias correction for cloud-free pixels was more effective for a deterministic forecast.

**Table 1: The biases of *O-B* for the selected case at 06:00 UTC on 1 September 2020. The comparison takes into account the sea and land surface types and cloudy and cloud-free scenarios. Here *N$_{match}$* denotes matched pixels in *O* and *B* for the selected cloud mask and surface type. *N$_{obs}$* denotes the number of pixels in *O* for the selected cloud mask and surface type. $\gamma$ denotes the bias correction coefficient.**

| Surface type | | Land | | Sea | | Land + Sea |
|---|---|---|---|---|---|---|
| cloud mask | | Cloud-free | Cloudy | Cloud-free | Cloudy | Cloud-free + Cloudy |
| | $N_{match}$ | 148046 | 1144077 | 43897 | 780237 | 2116257 |
| The | $N_{obs}$ | 198215 | 1836203 | 57091 | 132159 | 2223668 |
| deterministic | $\gamma$ | 0.0103 | 0.1753 | 0.1619 | 0.0524 | —— |
| forecast | uncorrected | 0.0014 | 0.1018 | 0.0065 | 0.0431 | 0.0634 |
| | corrected | -0.0002 | 0.0470 | 0.0000 | 0.0367 | 0.0330 |
| | $N_{match}$ | 68635 | 1675562 | 18488 | 1056007 | 2818692 |
| The | $N_{obs}$ | 205047 | 1846118 | 80850 | 1127039 | 3259054 |
| ensemble | $\gamma$ | 0.0144 | 0.3032 | 0.2694 | 0.1721 | —— |
| forecast | uncorrected | 0.0017 | 0.0989 | 0.0072 | 0.0705 | 0.0703 |

| | | Cloud-free | Cloudy | Cloud-free | Cloudy | Cloud-free + Cloudy |
|---|---|---|---|---|---|---|
| | corrected | -0.0004 | 0.0044 | 0.0001 | 0.0032 | 0.0010 |

**Table 2: Same as in Table 1, but the results are for the selected case at 06:00 UTC on 15 September 2020**

| Surface type | | Land | | Sea | | Land + Sea |
|---|---|---|---|---|---|---|
| cloud mask | | Cloud-free | Cloudy | Cloud-free | Cloudy | Cloud-free + Cloudy |
| | $N_{match}$ | 149281 | 1167745 | 27435 | 791246 | 2135707 |
| The | $N_{obs}$ | 164244 | 1910726 | 49734 | 1112529 | 3237233 |
| deterministic | $\gamma$ | -0.0180 | 0.1368 | -0.0128 | 0.1809 | —— |
| forecast | uncorrected | -0.0016 | 0.0955 | 0.0073 | 0.0353 | 0.0582 |
| | corrected | 0.0000 | 0.0517 | -0.0001 | 0.0389 | 0.0360 |
| | $N_{match}$ | 76933 | 1711970 | 7067 | 1051263 | 2847233 |
| The | $N_{obs}$ | 162862 | 1933710 | 56417 | 1107440 | 3260429 |
| ensemble | $\gamma$ | -0.0096 | 0.2862 | 0.1830 | 0.2216 | —— |
| forecast | uncorrected | -0.0017 | 0.0973 | 0.0075 | 0.0688 | 0.0501 |
| | corrected | -0.0003 | 0.0064 | 0.0001 | 0.0034 | -0.0050 |

The PDF of the $O$ - $B$ biases with and without bias correction were shown in Fig. 10. After bias correction, the right-side tail of the PDF for the $O$ - $B$ departure shrank, while opposite impact was introduced to the left side of the PDF. The results agree well with the fact that the $O$ - $B$ was positively biased for the selected cases. In addition, the PDFs for the $O$ - $B$ departure conformed to the Gaussian functions better for an ensemble forecast than a deterministic forecast. A potential explanation is that ensemble forecast is more effective to mitigate the random errors related to cloud simulations, which is a possible cause to the irregular structure of the PDF for a deterministic forecast. Therefore, the bias correction based on an ensemble forecast should increase the robustness of the bias correction method.

The pseudocode for the bias correction method based on the equivalents derived from ensemble forecasts was illustrated by the following:

**for** $m = 1: N_{obs}$ **do** ### $N_{obs}$ denotes the number of pixels in $O$

   **if** $sfc(m)$ → land **then** ### for land surface

      **if** $clm(m)$ → cloud-free **then** ### for cloud-free pixels in $O$

         $o'_m = o_m (1 - \gamma_{clr}^{land})$

      **else if** $clm(m)$ → cloudy **then** ### for cloudy pixels in $O$

         $o'_m = o_m (1 - \gamma_{cld}^{land})$

      **else:** ### for uncertain pixels in **O**

         $o'_m = o_m (1 - \gamma_{uct}^{land})$

**end if**

    **else if** *sfc(m)* → sea **then** ### for land surface

        **if** *clm(m)* → cloud-free **then** ### for cloud-free pixels in  *O*

$$o'_m = o_m(1 - \gamma^{sea}_{clr})$$

        **else if** *clm(m)* → cloudy **then** ### for cloudy pixels in  *O*

$$o'_m = o_m(1 - \gamma^{sea}_{cld})$$

        **else:**   ### for uncertain cloud mask in  *O*

$$o'_m = o_m(1 - \gamma^{sea}_{uct})$$

        **end if**

    **endif**

**end for**

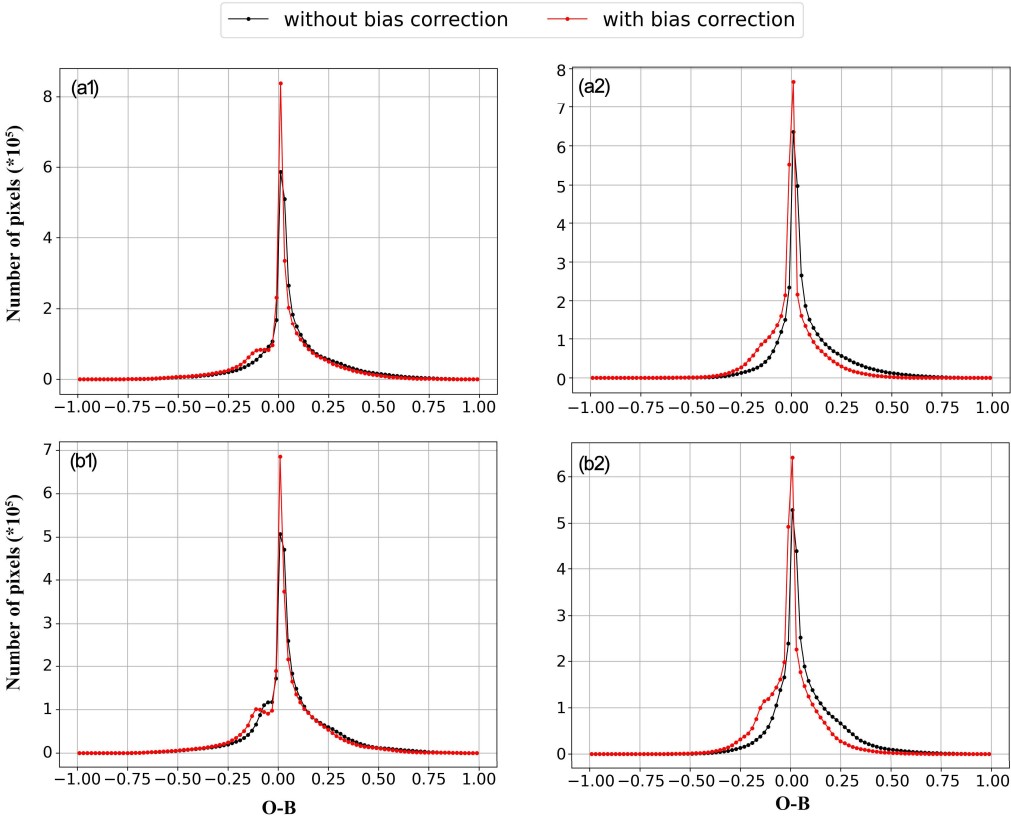

**Figure 10: The probability density distribution function of the *O-B* departure for FY-4A visible reflectance at 06:00 UTC on 15 September 2020 (a1-a2) and 17 September 2020 (b1-b2). From left to right, the two columns correspond to the results for deterministic forecasts (a1-b1) and ensemble forecasts (a2-b2), respectively.**

## 6. Conclusions

In this study, the FY-4A/AGRI channel 2 visible reflectance data were compared with the equivalents derived from the forecasts of the CMA-MESO model using the RTTOV-DOM forward operator. The spatiotemporal variations of the  *O - B*  biases were explored, and the main contributing factors to the  *O - B*  biases were discussed. In addition, a bias correction

method was suggested to correct the systematic biases of $O - B$, which will facilitate the data assimilation application of FY-4A/AGRI reflectance data. The main findings are summarized below.

Compared with $B$, $O$ was positively biased in most cases. The temporal variation characteristics of the $O - B$ biases revealed an abrupt change from 8$^{th}$ to 9$^{th}$ September 2020, when the calibration correction coefficients of FY-4A/AGRI channel 2 were updated by the NSMC. The $O - B$ biases were positively related to the domain-averaged observed reflectance, which confirmed that the abrupt change in the time series of $O - B$ biases for FY-4A was caused by the measurement calibration processes. The findings indicate that the reflectance derived from the CMA-MESO forecasts was capable of monitoring the performance of the FY-4A visible instrument, which was the normal routine for monitoring the infrared and microwave instruments in NSMC of China and other satellite instrument monitoring systems (Lu et al., 2020).

Apart from the measurement errors, the influences of forward-operator errors and NWP model errors were assessed by a series of sensitivity studies and synergic observations. Validation of the CMA-MESO forecasts by ground-based precipitation observations suggested general consistency between the CMA-MESO forecasts and observations, but the deficiencies of the model in simulating strongly convective weather system and reduced performance over complex terrain areas, as was suggested by previous studies, were confirmed. In addition, the forward-operator errors were especially evident for cloudy pixels since RTTOV-DOM suffers uncertainties in cloud optical properties. The impact of aerosols on $O$ was not considered in $B$ because CMA-MESO is unable to resolve aerosol processes currently. Sensitivity studies indicate that neglecting aerosols tends to decrease the TOA reflectance in cloud-free conditions. The impact of aerosols was complicated by aerosol type and aerosol vertical distribution.

Despite that the $O - B$ departure was collaboratively determined by many factors, systematic biases in $O - B$ were revealed, which facilitated the bias correction in data assimilation applications. Unlike the bias correction of infrared and microwave radiance data based on some well-designed predictors, the biases in visible reflectance data were simply corrected by the domain-averaged relative differences of $O - B$. The main reason is that the predictor-based bias correction could introduce extra errors in the background to the observations since the visible equivalents are largely influenced by uncertainties in background which are particularly evident in cloudy regions. The bias correction method was tested by two cases, and overall reduction of the biases was revealed. Since an ensemble forecast had advantages over a deterministic forecast in reducing the random errors of cloud simulations, the unbiased Gaussian distribution of $O - B$ departure was better respected for the ensemble-based bias correction.

It is noted that bias-corrected reflectance is largely determined by $B$. Despite that the representativeness errors in $B$ could be mitigated by more accurate forward operators (e.g. forward operators which accounts for 3D radiative effects) and more skilful NWP models (e.g., short-term forecasts based on advanced data assimilation method, ensemble forecasts which involve well-designed ensemble members, etc), $B$ derived from a deterministic forecast or an ensemble forecast will be inevitably associated with errors due to the deficiencies of both the NWP models and forward operators. Correcting the biases in $O$

based on *B* is a measure of last resort due to a lack of sufficient reference observations for comparing with the satellite observations. Whether the bias correction brings benefits to the numerical weather prediction should be tested by data assimilation in real-world whether systems. Extending the bias correction to data assimilation in real-world cases and sophisticated evaluation of data assimilation experiments are ongoing.

*Code availability.* Version 12.3 of RTTOV source code is publicly available at https://nwp-saf.eumetsat.int/site/software/rttov/rttov-v12/ (last access: 5 March 2019).

*Data availability.* The CMA-MESO short-term forecasts data in 2020 were provided by the CMA Earth System Modeling and Prediction Centre (CEMC). The 0.01°×0.01° multi-source observed precipitation products gridded were obtained from the National Meteorological Information Center (NMIC) of China Meteorological Administration (CMA) through the "Tianqing (天擎)" system. The 1km×1km FY-4A full-disk reflectance data, the 4km × 4km geometry (GEO) data and cloud mask products were obtained from the National Satellite Meteorological Center (NSMC) at http://satellite.nsmc.org.cn/PortalSite/Data/DataView.aspx?currentculture=zh-CN. The datasets are also available upon request from Yongbo Zhou (yongbo.zhou@nuist.edu.cn).

*Author contribution.* Yongbo Zhou devised the methodology, performed radiative transfer simulations using RTTOV, downloaded and processed the FY-4A/AGRI data, realised and evaluated the experiment, and wrote the paper. Yubao Liu supervised the research activity and provided the linux cluster for radiative transfer simulations and related calculations. Wei Han and HS provided the CMA-MESO short-term forecasts and some guidance on the processing of these data. Yuefei Zeng evaluated the experiment designs and revised the writing. All authors were involved in discussions throughout the development and experiment phase, and all authors commented on the paper.

*Competing interests.* The contact author has declared that none of the authors has any competing interests

*Acknowledgement.* We acknowledge the High Performance Computing Center of Nanjing University of Information Science & Technology for their support of this work. Yongbo Zhou would like to thank Dr. Zhanshan Ma from CMA Earth System Modeling and Prediction Centre and Dr. Daosheng Xu from Guangdong Provincial Key Laboratory of Regional Numerical Weather Prediction, CMA, for their help in understanding the CMA-MESO sub-grid processes. In addition, the authors thank two anonymous reviewers for their constructive comments on this manuscript, which are of great help in improving the quality of the manuscript.

*Financial support.* This research has been supported by the National Natural Science Foundation of China (No. 42305161, U2342222), the open topic grant funded by the Key Laboratory of high Impact Weather (special), China Meteorological Administration, and the Natural Science Foundation of Jiangsu Province (No. BK0210665).

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
