# Peer review of "Exploring the characteristics of FY-4A/AGRI visible reflectance using the CMA-MESO forecasts and its implications to data assimilation"

_Atmospheric Measurement Techniques, 2024_

## Author Comment (AC1)

**Response to reviewer**

- **General comment:**

This paper compares the FY-4A/AGRI 0.65-um visible reflectance (O) with the model simulations generated from CMA-MESO forecasts using the RTTOV (B). The potential sources contributing to the differences between O and B, such as the unresolved aerosol processes, the ice scattering models, are analyzed.

The paper is relevant to the cloud remote sensing field, as the growing international fleet of next-generation geostationary imagers can be expected to aid in our understanding of the diurnal cycles of clouds and aerosols. Well understood and characterized the biases of their observations will therefore be well received by the community. However, the authors make what I think are several unsubstantiated assertions (see my detailed comments). I recommend major revisions before reconsidering for publication. My general and specific comments are below.

- **Our reply to general comment:**

The authors thank the anonymous reviewer for the constructive comments. We made major revisions to the manuscript, including the evaluation of the forecasts of CMA-MESO, discussions on the spatial and seasonal variations of the O-B biases, corrections on some typo errors (e.g., abbreviations, inappropriate usage of "evaluation", etc.). As a result, the outline of the manuscript was changed compared with the initial version. Some of the revisions were made according to the reviewer's comments. Some were made by the authors spontaneously to improve the readability. The point-by-point response to the reviewer's comments were provided below.

- **General Comment 1:**

A comparison with the model simulations cannot be called an "evaluation", especially when the model simulations are not as accurate as expected. Currently, the RTTOV forward-operator for clouds and/or within the visible and shortwave infrared spectral ranges is still questionable, and the forecasts from CMA-MESO also lack adequate evaluations.

- **Our reply to general comment 1:**

Thank you for pointing this out. It is true that the reflectance simulated from the forecasts of CMA-MESO model using RTTOV cannot represent the true reflectance due to the deficiencies of the CMA-MESO and RTTOV models. Therefore, the title of this manuscript was changed to "Exploring the characteristics of FY-4A/AGRI visible reflectance using the forecasts of the CMA-MESO model and its implications to data assimilation". (L1-3 in the revised manuscript)

- **General Comment 2:**

As (1), if the authors persist in characterizing the biases of AGRI reflectance observations by comparing with the model simulations, the performances of RTTOV forward-operator and the forecasts from CMA-MESO should be evaluated first.

- **Our reply to general comment 2:**

We agree with the reviewer that the evaluation of the CMA-MESO model and the RTTOV-DOM forward operator is necessary when addressing the O (FY-4A/AGRI 0.65-um visible reflectance) - B (model simulations generated from CMA-MESO forecasts using the RTTOV) differences. Accordingly, two main revisions were made to address this problem.

(1) Evaluation of the forecasts of CMA-MESO

The forecasts of CMA-MESO model were evaluated by comprehensive observations.

Firstly, the cloud mask diagnosed from the forecasts of the CMA-MESO model was compared with the spatiotemporally collocated cloud mask products derived from the Himawari-8 geostationary satellite. Quantitative analysis indicated a Fraction Skill Score (FSS) larger than 0.7 in most cases (Fig. 5 in the revised manuscript, L232-234), implying that the CMA-MESO model performs well in forecasting cloud locations.

[Figure]

**Figure 5: Fraction Skill Scores (FSSs) of cloud cover for the short-term forecasts of CMA-MESO with different forecasting lead times. The FSSs were calculated with a square of length of 2.**

Secondly, the one-hour accumulated precipitation was compared with the observations provided by the multi-source observed precipitation products in Chinese mainland. The results indicated that the CMA-MESO model could reproduce the precipitating areas with adequate accuracy (Fig. 6 in the revised manuscript, L243-247). Quantitative analysis of the Enhanced Threatening Score (ETS), a commonly used metrics for evaluating the accuracy of precipitation forecasting, revealed better performance of CMA-MESO in forecasting the light to moderate rain (Fig. 7 in the revised manuscript, L248-251). Despite that some spatial discrepancies of the core of heavy rain were revealed between the forecasts and observations, the domain-averaged precipitation agrees well with the observation (Fig. 8 in the revised manuscript, L263-265). The results imply that the forecasts of CMA-MESO should have certain reliability, especially when it comes to the domain-averaged quantities (remember that the O-B biases were calculated on the domain average).

[Figure]

**Figure 6: Accumulated one-hour precipitation at 06:00 UTC derived from the multi-source observed precipitation products (the first column), the 6-h forecasts of CMA-MESO (the second column), and the ensemble forecasts including seven ensemble members which have different forecasting lead times. From top to bottom, each panel corresponds to the results on 2 September 2020 (a1-a3), 8 September 2020 (b1-b3), 14 September 2020 (c1-c3), 20 September 2020 (d1-d3), and 26 September 2020 (e1-e3).**

[Figure]

**Figure 7: Enhanced Threatening Score (ETS) for the accumulated one-hour precipitation at 06:00 UTC derived from the forecasts of CMA-MESO with different forecasting lead times. From top to bottom, each precipitation range represents the (a) light rain, (b) moderate rain, (c) heavy rain, (c) rainstorm, and (e) heavy downpour. Fcst-03 means the 3-h forecasts at 06:00 UTC, and so forth.**

[Figure]

**Figure 8: Domain-averaged one-hour accumulated precipitation at 06:00 UTC derived from the short-term forecasts of CMA-MESO with different forecasting lead times. Fcst-03 means the 3-h forecasts at 06:00 UTC, and so forth.**

The evaluation using two observations both revealed the advantages of an ensemble forecast over a deterministic forecast in reducing the errors in cloud simulations. The implication to the bias correction of data assimilation is that the B derived from an ensemble forecast should be a better choice than from a deterministic forecast.

(2) Discussions on the deficiencies of RTTOV (L266-294)

We feel helpless to evaluate the performance of the RTTOV model. The largest challenge comes from a lack of accurate observations corresponding to the real atmosphere state variables. Therefore, some of the major deficiencies were discussed in the revised manuscript.

Although great strides have been made to RTTOV, we have to admit that the simulations of RTTOV in cloudy regions are still questionable. Potential errors include the ignored 3D radiative effects, the uncertainties of the cloud optical properties in the build-in cloud schemes, etc. For example, the pre-assumed cloud particle size distribution (PSD) inherent in the cloud schemes in RTTOV is inconsistent with that of NWP models, not to mention the representativeness of the pre-assumed PSD in real cases. To illustrate this problem, a sensitivity study was performed by RTTOV configured with two different ice schemes, i.e., the Baum and Baran schemes. Distinct differences were revealed for the simulated reflectance (Fig. in the revised manuscript, L287-289), which confirms the uncertainties in the cloud optical properties of the RTTOV model.

[Figure]

**Figure 9: The impact of ice cloud schemes on the TOA reflectance. $r_{ref}$ denotes the reflectance simulated by RTTOV configured with the ice scheme of Baran et al. (2015). $r_{ice}$ denotes the simulations based on the SSEC/Baum ice scheme.**

To the best of our knowledge, we do not know the actual representativeness of the optical properties derived from the built-in cloud schemes. Sophisticated evaluation will be needed to address the performance these build-in cloud schemes in real cases.

Currently, B derived from any of the NWP models and forward operators will enviably suffer from errors. As long as B is systematically biased, correcting the biases in O based on B and other predictors, which is the routine operation for the data assimilation of infrared and microwave satellite observations, will be questionable. Therefore, we have to admit that the O-B method is a measure of last resort due to a lack of sufficient reference observations for comparing with the satellite observations. Whether the bias correction brings benefits to the numerical weather prediction should be tested by data assimilation in real-world whether systems and should be evaluated by comprehensive observations.

- **General Comment 3:**

    The bias characteristics are not well analyzed. (1) How about the spatial distributions or seasonal variations of AGRI biases? (2) Do they have differences before and after the FY-4A satellite's U–turn at the vernal and autumnal equinoxes?

- **Our reply to general comment 3:**

    (1) Spatiotemporal variation of the O-B biases (L178-211)

    To better characterizing the spatiotemporal variation of the O-B biases, extra simulations were performed for March, June, and December. We did not perform the simulations for the whole 2020 year mainly because the radiative transfer simulation is rather computationally comprehensive. On our linux cluster which is equipped with 2.20 GHz Xeon Silver 4214 CPU, it will take approximately 30 min ~ 1 hour (32 CPUs for parallel computation) for the RTTOV-DOM (V12.3) to generate a synthetic visible imagery which includes 2501×1671 pixels (the CMA-MESO grids). We think the results for March, June, September, and December could reveal some seasonal variation characteristics of the O-B biases.

Based on the four-month simulations over the CMA-MESO domain, the temporal (Fig. 3 in the revised manuscript, L193-196) and spatial (Fig. 4 in the revised manuscript, L209-211) variation characteristics of the O-B biases were explored. The results indicate different spatiotemporal variations of the O-B biases for the four months, which is closely related to the spatiotemporal of the performance of the CMA-MEOS model, the variation of aerosol properties, etc.

[Figure]

**Figure 3: Time series of the O-B biases for the cloud-free, cloudy, and all pixels for FY-4A visible observations in (a) March, (b) June, (c) September, and (d) December. The time series of the O-B biases for Himawari-8 visible observations for all pixels was also provided in (c) for comparison with that of FY-4A.**

[Figure]

**Figure 4: Spatial distribution of the O-B biases for FY-4A visible observations in (a) March, (b) June, (c) September, and (d) December.**

(2) Characteristics during U–turn at the vernal and autumnal equinoxes (Fig. 3, L193-196)

In the Northern Hemisphere, the vernal and autumnal equinoxes falls about 20 March and September 22 or 23, respectively. During this period of time, the Sun crosses the celestial equator, leading to changes in the sun-satellite geometries. Checking through the time series of the O-B biases in March and September, we see no differences during these two days or around when compared with other dates (Fig. 3(a) and Fig. 3(c) in the revised manuscript, L193-196). Therefore, a tentative conclusion could be drawn that the temporal variations of the O-B biases do not have differences before and after the FY-4A satellite's U–turn at the vernal and autumnal equinoxes.

- **Specific Comment 1:**

Lines 16, 22, 33 and 72: The abbreviations (FY, TOVS, and so on) should be given full name when first appeared in the abstract and text.

- **Our response to specific comment 1:**

The abbreviations were checked throughout the revised manuscript, and the full names were given the first time they appeared in the abstract and text. E.g., FY is the abbreviation of "Fengyun", TOVS is the abbreviation of Television infrared observation satellite Operational Vertical Sounder, and RTTOV is the abbreviation of Radiative Transfer for the Television infrared observation satellite Operational Vertical Sounder, etc. (L17, L23-24, L35, L76-77, etc. in the revised manuscript)

- **Specific Comment 2:**

Line 85: Himawari-8 satellite should be introduced because not all readers know it is the first one of the Japanese next-generation geostationary satellite.

- **Our response to specific comment 2:**

In the revised manuscript, we added some introductions to the Himiwari-8 satellite. (L106-107 and L152-164 in the revised manuscript)

"…… Himawari-8, the first one of the Japanese next-generation geostationary satellite ……" (L106-107)

Himawari-8 was launched by the Japan Meteorological Agency on October 7, 2014. The geostationary satellite carries the Advanced Himawari Imager (AHI) which provides radiance observations covering visible to infrared spectra and completes a full-disk scan every 10 minutes. The Himawari-8/AHI band 3 (0.55 μm - 0.72 μm) is close to the FY-4A/AGRI channel 2, which contains critical information on clouds, aerosols, and underlying surfaces (Bessho et al., 2016). In this study, the Himawari-8 cloud mask products gridded at 5 km resolution in September 2020

were used to evaluate the performance of the CMA-MESO in predicting cloud locations. To ensure the spatial collocation between the observations and simulations, the Himawari-8 cloud mask products were interpolated to the CMA-MESO grids by a bi-linear interpolation. In addition, the Himawari-8 reflectance data gridded at 5 km $\times$ 5 km resolution in September 2020 were used to explore the stability of Himiwari-8 visible observations. The observed reflectance data were interpolated to the CMA-MESO grids to ensure spatial collocation with the simulated equivalents. Since the synthetic imageries for the FY-4A and Himaeari-8 were derived from the same forecasts from CMA-MESO, the differences in the time series of O-B biases for the two satellites should reveal some different characteristics of their corresponding visible instruments. (L152-164)

- **Specific Comment 3:**

  Line 82: How about the spatial coverage of CMA-MESO, or the region of interest in this study?

- **Our response to specific comment 3:**

  In the revised manuscript, the spatial coverage of CMA-MESO was shown by Fig. 1, which is also the region of interest in this study. (L95-97 in the revised manuscript)

[Figure]

**Figure 1: The domain coverage of the CMA-MESO model, which includes 2501×1671 horizontal grids with a horizontal grid spacing of 0.03 °.**

- **Specific Comment 4:**

  Lines 96 and 117? Here, the authors give two cloud mask definitions. Which one will be used for Tables 1 and 2?

- **Our response to specific comment 4:**

  We are sorry for not making it clear here. The two cloud masks are defined for the synthetic imageries (observations, O) and observed imageries (simulations, B) separately. For the observed imageries, cloud masks were directly derived from the cloud mask products. For the synthetic imageries, cloud masks were dragonized from the CWP with a threshold value of 0.01 kg m$^{-2}$.

  For spatiotemporally collocated observations and simulations, the O-B biases were calculated for the pixels which are designated to be cloudy and cloud-free for both O and B. The O-B biases for the cloudy and cloud-free pixels were further used to correct the systematic biases of the corresponding scenarios separately. (L136-139 in the revised manuscript)

- **Specific Comment 5:**

  Lines 201-203: I can't understand this sentence. Aren't the "microphysical properties therein" "cloud variables"?

- **Our response to specific comment 5:**

  We were intended to say that "Compared with the infrared and microwave radiance observations, the visible reflectance is much more sensitive to cloud variables, regardless of the type of cloud hydrometeors or the vertical location of clouds. In contrast, the infrared radiance data are only sensitive to cloud-top properties due to strong absorption effects of clouds in infrared spectra".

  This part was revised in the revised manuscript. (L334-336 in the revised manuscript)

- **Specific Comment 6:**

  Figure 6: Readers can hardly identify the differences between observed and model simulated reflectance. The authors are suggested using a different colormap or adding figures to show their differences.

- **Our response to specific comment 6:**

  This figure was modified in the revised manuscript. The first and second columns shows the observed and simulated imageries. They share the same colorbar to make a fair comparison between them. The third column shows the O-B departure of the imageries. We think the revised figure could depict the differences between O and B more clearly. (L367-370 in the revised manuscript)

[Figure]

**Figure 11: The observed and bias-corrected reflectance at 06:00 UTC on 1 September 2020 (a1-a3) and 15 September 2020 (b1-b3). From left to right, the three columns correspond to the observed imageries (a1-b1), the ensemble mean synthetic imageries (a2-b2), and the observation-minus-simulation imageries (a3-b3).**

---

## Author Comment (AC2)

**Response to reviewer #1**

We received another reviewer's comments on this manuscript, and extra modifications were made. Therefore, this version of response letter has some differences compared with the previously uploaded one, but your valuable suggestions were respected and the revisions were made according to your general and specific comments. Please take this version as the final confirmation. We are sorry for the inconvenience.

**General comment:**

This paper compares the FY-4A/AGRI 0.65-um visible reflectance (O) with the model simulations generated from CMA-MESO forecasts using the RTTOV (B). The potential sources contributing to the differences between O and B, such as the unresolved aerosol processes, the ice scattering models, are analyzed.

The paper is relevant to the cloud remote sensing field, as the growing international fleet of next-generation geostationary imagers can be expected to aid in our understanding of the diurnal cycles of clouds and aerosols. Well understood and characterized the biases of their observations will therefore be well received by the community. However, the authors make what I think are several unsubstantiated assertions (see my detailed comments). I recommend major revisions before reconsidering for publication. My general and specific comments are below.

**Our response:**

The authors thank the anonymous reviewer for the constructive comments. We made major revisions to the manuscript, including the evaluation of the forecasts of CMA-MESO, discussions on the spatial and seasonal variations of the O-B biases, corrections on some typo errors (e.g., abbreviations, inappropriate usage of "evaluation", etc.). As a result, the outline of the manuscript was changed compared with the initial version. Some of the revisions were made according to your valuable comments. Some were made according to

the comments of another reviewer. The point-by-point response to the reviewer's comments were provided below.

**General Comment 1:**

A comparison with the model simulations cannot be called an "evaluation", especially when the model simulations are not as accurate as expected. Currently, the RTTOV forward-operator for clouds and/or within the visible and shortwave infrared spectral ranges is still questionable, and the forecasts from CMA-MESO also lack adequate evaluations.

**Our response:**

Thank you for pointing this out. It is true that the reflectance simulated from the forecasts of CMA-MESO model using RTTOV cannot represent the true reflectance due to the deficiencies of the CMA-MESO and RTTOV models. Therefore, the title of this manuscript was changed to "Exploring the characteristics of FY-4A/AGRI visible reflectance using the CMA-MESO forecasts and its implications to data assimilation". (L1-2)

**General Comment 2:**

As (1), if the authors persist in characterizing the biases of AGRI reflectance observations by comparing with the model simulations, the performances of RTTOV forward-operator and the forecasts from CMA-MESO should be evaluated first.

**Our response:**

We agree with you that the evaluation of the CMA-MESO model and the RTTOV-DOM forward operator is necessary when addressing the O (FY-4A/AGRI visible reflectance) - B (model simulations generated from CMA-MESO forecasts using the RTTOV) differences. Accordingly, two major revisions were made.

(1) Evaluation of the forecasts of CMA-MESO (L195-227)

Firstly, the one-hour accumulated precipitation was compared with the observations provided by the multi-source observed precipitation products in Chinese mainland. Good agreement between the simulations and observations was revealed, except that the precipitation areas were overestimated by the CMA-MESO forecasts in Chinese mainland. In addition, the precipitation was overestimated by the CMA-MESO forecasts. The evaluation results suggest that despite general agreement between the observations and simulations were revealed, the CMA-MESO forecasts suffer from deficiencies especially over complex terrain areas.

Secondly, the Probability density Distribution Functions (PDFs) of one-month Brightness Temperature (BT) for the FY-4A/AGRI channel 13 (10.30 μm – 11.30 μm) observations and simulations was analyzed. The results were shown in Fig. 4. The PDF was underestimated at the high-BT end. In contrast, it was overestimated at the low-BT end. Since channel 13 is an infrared window channel, BT in cloudy areas is directly related to cloud top height. Therefore, the PDF analysis implies that high-level clouds were underestimated by CMA-MESO whereas low-level clouds were overestimated. The evaluation suggested deficiencies of the CMA-MESO model in forecasting high-level clouds.

(2) Discussions on the uncertainties of RTTOV (L66-80, L341-385)

We feel helpless to evaluate the performance of the RTTOV model. The largest challenge comes from a lack of accurate observation of reflectance corresponding to the real atmosphere state variables. Therefore, instead of evaluating the performance of RTTOV by comprehensive observations, the performance of RTTOV and the uncertainties of the forward operator were discussed in the revised manuscript.

RTTOV was widely used to generate synthetic visible images. The synthetic images were further compared with satellite observed visible images to better understand the observation errors and representativeness errors and to provide guidance for the improvements of NWP models and forward operators. To save computational cost, a method for fast satellite image synthesis (MFASIS) was developed based on a lookup table (LUT) computed with one-dimensional (1D) solver of RTTOV in rotated Cartesian coordinates to account for some three-dimensional (3D) radiative effects (Scheck et al., 2016; Scheck et al., 2018). Intercomparison of satellite visible reflectance and the equivalents derived from NWP models and MFASIS indicated generally good agreement, and the Bidirectional Reflectance Distribution Function (BRDF) of land surface derived from a monthly mean atlas generated reasonable results in cloud-free conditions (Lopez

and Matricardi, 2022). Data assimilation of satellite visible reflectance data based on the MFASIS suggested positive impacts in real-world cases (Scheck et al., 2020). Since March 2023, satellite visible reflectance data have been operationally assimilated in German Weather Service by using the MFASIS forward operator. Although most of the studies are based on the MFASIS solver, the error estimates derived for MFASIS present upper bounds for RTTOV-DOM since the latter is just an emulator for the latter used in this study. Therefore, it is expected that RTTOV could generate reliable visible images if the NWP models were well tuned and the configurations of RTTOV were optimized.

However, knowledge on the cloud optical properties is scarce, which may introduce some uncertainties to the simulated reflectance. For example, the pre-assumed cloud particle size distribution (PSD) inherent in the cloud schemes in RTTOV is inconsistent with that of NWP models, not to mention the representativeness of the pre-assumed PSD in real cases. To illustrate this problem, a sensitivity study was performed by RTTOV configured with two different ice schemes, i.e., the Baum and Baran schemes. Distinct differences were revealed for the simulated reflectance, which confirms the uncertainties in the cloud optical properties of the RTTOV model.

According to the evaluation of the forecasts of CMA-MESO and discussions on the uncertainties of RTTOV, B derived from the CMA-MESO+RTTOV similations will enviably suffer from errors. Therefore, we have to admit that the O-B method is a measure of last resort due to a lack of sufficient reference observations for comparing with the satellite observations. Whether the bias correction brings benefits to the numerical weather prediction should be tested by data assimilation in real-world whether systems and should be evaluated by comprehensive observations.

**General Comment 3:**

The bias characteristics are not well analyzed. (1) How about the spatial distributions or seasonal variations of AGRI biases? (2) Do they have differences before and after the FY-4A satellite's U–turn at the vernal and autumnal equinoxes?

**Our response:**

(1) Spatiotemporal variation of the O-B biases (L228-261)

To better characterizing the spatiotemporal variation of the O-B biases, extra simulations were performed for March, June, and December. We did not perform the simulations for the whole 2020 year mainly because the radiative transfer simulation is rather computationally expensive. On our linux cluster which is equipped with 2.20 GHz Xeon Silver 4214 CPU, it will take approximately 30 min ~ 1 hour (32 CPUs for parallel computation) for the RTTOV-DOM (V12.3) to generate a synthetic visible imagery which includes 2501×1671 pixels (the CMA-MESO grids). We think the results for March, June, September, and December could reveal some seasonal variation characteristics of the O-B biases.

For the spatial distribution of O-B biases, systematic biases were revealed over the Southern foothills of the Himalayas, the Sichuan basin, and the Yunnan-Kweichow Plateau, both in September (Fig. 5, L255) and in other months (Fig. S2 in the supplementary material). On one hand, some areas of the Qinghai-Tibet Plateau were covered with snow. Reflectance simulated in these areas should be less accurate compared with other places since the BRDF atlas is questionable in snow-covered areas (Ji et al., 2022). On the other hand, the performance of the CMA-MESO model was reduced over complex terrain areas. The analysis of the spatial distribution of O-B biases suggested that the snow-covered areas and complex terrain areas should be excluded in the following analysis, mainly because that it is clear that one cannot get reasonable results in these areas.

Based on the four-month simulations over the CMA-MESO domain, the spatial (Fig. 5 for the September (L255) and Fig. S2 for March, June and December in the supplementary material) and temporal (Fig. 7 for the September (L331) and Fig. S3-Fig. S5 for March, June and December in the supplementary material) variation characteristics of the O-B biases were explored. The results indicate different spatiotemporal variations of the O-B biases for the four months, which is closely related to the spatiotemporal of the performance of the CMA-MEOS model, the variation of aerosol properties, etc.

(2) Characteristics during U–turn at the vernal and autumnal equinoxes (Fig. 7 in the revised manuscript (L331) and Fig. S3 in supplementary material)

In the Northern Hemisphere, the vernal and autumnal equinoxes falls about 20 March and September 22 or 23, respectively. During this period of time, the Sun crosses the celestial equator, leading to changes in the sun-satellite geometries. Checking through the time series of the O-B biases in March and September, we see no differences during these two days or around when compared with other dates. Therefore, a tentative conclusion could be drawn that the temporal variations of the O-B biases do not have differences before and after the FY-4A satellite's U–turn at the vernal and autumnal equinoxes.

Nevertheless, we found an interesting phenomenon for the temporal variation of the O-B biased in June. An abrupt change was revealed on June 21th (Fig. S4(b)). The abrupt change was caused by the annular solar eclipse on 06:00 UTC 21 June 2020, when the incoming solar radiance was sheltered by the moon over the west parts of the CMA-MESO domain. The annular solar eclipse caused an abrupt decrease of the photons received by the AGRI visible channel. As a result, the visible image was darkened. The darkened visible image was also revealed by the National Aeronautics and Space Administration (NASA) worldview project (https://worldview.earthdata.nasa.gov/). However, the annular solar eclipse was not considered when performing the radiative transfer simulations by RTTOV-DOM. Instead, the incoming solar irradiance was set to a constant, which caused an abrupt decrease of the O-B biased. (L324-330)

**Specific Comment 1:**

Lines 16, 22, 33 and 72: The abbreviations (FY, TOVS, and so on) should be given full name when first appeared in the abstract and text.

**Our response:**

The abbreviations were checked throughout the revised manuscript, and the full names were given the first time they appeared in the abstract and text. E.g., FY is the abbreviation of "Fengyun" (L17, L35), and RTTOV is the abbreviation of Radiative Transfer for the Television infrared observation satellite Operational Vertical Sounder, etc. (L67-68, etc.)

**Specific Comment 2:**

Line 85: Himawari-8 satellite should be introduced because not all readers know it is the first one of the Japanese next-generation geostationary satellite.

**Our response:**

Corrected. (L135-136)

**Specific Comment 3:**

Line 82: How about the spatial coverage of CMA-MESO, or the region of interest in this study?

**Our response:**

In the revised manuscript, the spatial coverage of CMA-MESO was shown by Fig. 1, which is also the region of interest in this study. (L114-116)

**Specific Comment 4:**

Lines 96 and 117? Here, the authors give two cloud mask definitions. Which one will be used for Tables 1 and 2?

**Our response:**

We are sorry for not making it clear here. In the original manuscript, two cloud masks are defined for the synthetic images (observations, O) and observed images (simulations, B) separately. For the observed images, cloud masks were directly derived from the cloud mask products. For the synthetic imageries, cloud masks were dragonized from the CWP with a threshold value of 0.01 kg m$^{-2}$. For spatiotemporally collocated observations and simulations, the O-B biases were calculated for the pixels which are designated to be cloudy and cloud-free for both O and B. The O-B biases for the cloudy and cloud-free pixels were further used to correct the systematic biases of the corresponding scenarios separately.

According to another reviewer's comment, different definitions of cloud mask for observations and simulations could introduce mismatch of cloudy or cloud-free scenarios in the observed and simulated visible images. Therefore, an equivalent criterion of cloud mask for observed and simulated images was introduced to the revised manuscript. In the

revised manuscript, cloud mask was determined by comparing the simulated and observed reflectance with the reflectance simulated by ignoring cloud impacts. (L278-303)

For the synthetic visible image, a pixel was designated to be cloudy if the simulated reflectance $r_{sim}$ satisfies Equation (4). Otherwise, the pixel would be classified to be cloud-free.

$$r_{sim} > r_{sim,clear} \tag{4}$$

where $r_{sim,clear}$ denotes the simulated reflectance when cloud contributions were ignored.

The aerosol contributions were neglected by the simulations since the CMA-MESO forecasts do not provide aerosol information explicitly, whereas the observed reflectance inevitably includes aerosol contributions. Considering the aerosol contributions to the reflectance, a pixel is assumed to be cloudy if the observed reflectance $r_{obs}$ satisfies Equation (5),

$$r_{obs} > r_{sim,clear} + r_{aer}^{75} \tag{5}$$

where $r_{aer}^{75}$ denotes the aerosol contributions to the reflectance of cloud-free pixels, which was set to the upper quartile of $r_{obs,clear} - r_{sim,clear}$ for the preliminarily estimated cloud-free pixels. $r_{obs,clear}$ denotes the observed reflectance for cloud-free pixels, which were preliminarily determined by the FY-4A CLM product. The second-step estimate of cloud-free pixels was determined Equation (6),

$$r_{obs} < r_{sim,clear} + r_{aer}^{25} \tag{6}$$

where $r_{aer}^{25}$ denotes the aerosol contributions to the cloud-free reflectance. Similarly, $r_{aer}^{25}$ was set to the lower quartile of $r_{obs,clear} - r_{sim,clear}$ for the preliminarily estimated cloud-free pixels. The two-step estimate of cloud mask for observed images was performed to maintain equivalent criterion of the cloud mask for synthetic images. It is noted that the first-step estimate of cloud mask should have different representativeness compared with the cloud mask diagnosed from Equation (4). For example, the CLM cloud mask was generated by including extra infrared observations (Wang et al., 2019) that are much more sensitive to optically thin cloud, which may appear to be transparent in the visible band. Nevertheless, the quartile estimation should mitigate the impacts. On one hand, thin clouds which are transparent in the visible channel whereas are opaque in the infrared channels should contribute insignificant part to $r_{obs}$. On the other hand, the quartile estimation in Equations (4) and (5) discarded 25% samples in estimating the lower and upper quartiles of $r_{obs,clear} - r_{sim,clear}$.

**Specific Comment 5:**

Lines 201-203: I can't understand this sentence. Aren't the "microphysical properties therein" "cloud variables"?

**Our response:**

We were intended to say that "Compared with the infrared and microwave radiance observations, the visible reflectance is much more sensitive to cloud variables, regardless of the type of cloud hydrometeors or the vertical location of clouds. In contrast, the infrared radiance data are only sensitive to cloud-top properties due to strong absorption effects (Li et al., 2022)". This part was revised in the revised manuscript (L422-424)

**Specific Comment 6:**

Figure 6: Readers can hardly identify the differences between observed and model simulated reflectance. The authors are suggested using a different colormap or adding figures to show their differences.

**Our response:**

Thanks for pointing this out. Since O-B is positively biased for the selected cases, reflectance of the bias-corrected visible image should be reduced by a factor of $\gamma$, which denotes the bias correction coefficient (Equation (4), L435). The bias-corrected visible image remains general characteristics of the original manuscript. As a result, the contour maps of the original image and bias-corrected image would be rather similar, except that the bias-corrected image was slightly darker than the original one. Therefore, it is difficult to differentiate the original and bias-corrected visible images in contour maps, and the contour maps in the original manuscript were deleted in the revised manuscript.

Instead of presenting the results by contour maps, the O-B biases with and without bias correction were summarized in Table 1 (based on deterministic forecasts) and Table 2 (based on ensemble forecasts). The results should be shown in a clearer way in the revised manuscript. (L454-458)

---

## Author Comment (AC3)

**Response to reviewer #2**

**General Comments:**

In this manuscript, differences between observed visible satellite images from the AGRI instrument onboard FY-4A and synthetic images computed with the RTTOV forward operator package from deterministic forecasts of the CMA-MESO model are discussed and a bias correction method is proposed. The authors address a relevant question. Fast forward operators have become available for the up to now underused visible satellite channels in the last years, several services have started monitoring experiments for these channels and at the German weather service visible reflectances are assimilated operationally since March 2023. So there is a clear interest in exploiting the cloud and aerosol information contained in visible channels for data assimilation and understanding systematic errors is an important first step.

While the topic is clearly relevant, the methods employed by the authors are in my opinion not really sufficient to get information on the origin of reflectance errors. The latter could be caused by the instrument (e.g. calibration problems), the model (e.g. deficiencies in the representation of clouds) or the forward operator (e.g. albedo errors or 3D effects). In particular, I think the authors skipped one first, important step: Comparing reflectances histograms for O and B. In contrast to O-B statistics, histograms are not sensitive to the location of the clouds (to correct that is the job of data assimilation) and can provide a lot of information (see e.g. Geiss et al 2021). I would also suggest to distinguish not only between cloudy and clear pixels, but also between regions for which we would expect different error characteristics. Given the CMA-MESO domain contains the Himalaya and oceans, it could make sense to distinguish between land, sea (lower albedo error) and extreme orography (potentially larger model and albedo errors). And finally, it would be important to exclude cases from the statistics, for which it is clear that one cannot get reasonable results. This would e.g. be pixel for which the BRDF atlas contains contribution from snow and thus no reliable albedo information is available.

The bias correction proposed in the last part of the manuscript is in principle interesting. However, the results are not discussed in a sufficiently clear way and I do not understand the motivation for the "ensemble" version.

**Our response:**

Thank you for the instructive comments. According to your comments, major revisions were made to the manuscript, and the revisions include the following three main aspects.

(1) We added the analyses of the Probability distribution Density Functions (PDFs) for the reflectance and brightness temperature in the revised manuscript. (L200-216, L234-261)

The brightness temperature was analyzed for the FY-4A/AGRI channel 13 (10.30 μm – 11.30 μm), which is an infrared window channel. Therefore, the brightness temperature could reflect cloud top height in cloudy regions. For the BT images generated by the CMA-MESO+RTTOV-DOM simulations, the PDF was underestimated at the high-BT end and overestimated at the low-BT end, implying that high clouds were underestimated or low clouds were overestimated. For the visible images generated by the CMA-MESO+RTTOV-DOM simulations, the low-reflectance end of the PDF was overestimated, which should be related to the underestimated cloud cover and neglected aerosol contributions. In the medium reflectance range, the PDF was underestimated by the synthetic visible images, which should be related to the deficiencies of the NWP models and the forward operators (potential biases in the cloud optical properties).

(2) In the revised manuscript, the O-B biases were explored for different scenarios by distinguishing not only between cloudy and cloud-free pixels, but also between sea and land surfaces. In addition, three sets of threshold tests were applied to exclude cases where reasonable results cannot be expected. (L262-334)

The first threshold test is terrain height test. Due to the deficiencies of the CMA-MESO model over the complex terrain areas, especially over the Qinghai-Tibet Plateau, a threshold test of terrain height (> 4.0 km) was applied to exclude these areas.

The second threshold test is the land surface albedo test. Over land, the snow-covered areas were screened out by applying a threshold test of the surface BRDF (0.2/3.14). Here 0.2 is the surface albedo of typical snow-covered surfaces, and 0.2/3.14 denotes the BRDF for a Lambertian radiator

The third threshold test is the cloud-free reflectance test over sea areas. The highly reflective areas over ocean surface (reflectance > 0.1) were excluded to reduce the sun-glint impacts.

(3) For the data assimilation of FY-4A visible reflectance data in real-world cases, it is necessary to correct the systematic biases in the observations in order to comply with the unbiased Gaussian PDF of the observation errors. By applying the first-order bias correction with a deterministic forecast, the bias correction was not that effective since the Gaussianness of the PDF was not well respected. In fact, some irregular structures were revealed in the PDF. We think an important reason is that the synthetic images were accompanied with some random errors in the deterministic forecasts from CMA-MESO. An ensemble forecast should mitigate some of the random errors in the synthetic images. After bias correction based on an ensemble forecast, the Gaussianness of the PDF for O-B differences were better respected, and the biases were reduced more evidently compared with the bias-corrected results based on a deterministic forecast. This is the reason why an ensemble forecast was introduced in comparison with the results with a deterministic forecast. (L417-470)

**Specific Comment:**

- l. 24: How do you mean "negatively biased"? In Fig. 2 and Tab. 1 O-B is in general positive.

**Our response:**

We made a typo error in the original manuscript. This mistake was corrected in the revised manuscript. (L25-26)

**Specific Comment:**

- l. 27: The standard deviations are actually not reduced significantly and that is also not what you would expect from a bias correction

**Our response:**

It is true that the effects of the bias correction method on the standard deviations were trivial. As a result, discussions on the standard deviations were deleted in the revised manuscript.

**Specific Comment:**

- Introduction: There are two papers comparing synthetic and observed visible reflectances that should be cited here, Geiss et al. (2021) and Lopez & Matricardi (2022).

**Our response:**

Thank you for recommending the two papers which are closely related to the topic in our manuscript. The two papers were cited in the introduction in other parts of the manuscript. (L72-76)

**Specific Comment:**

- l. 57: "The key assumption [...] is that the model equivalents do not generate systematic biases" Well, this is just not a reasonable assumption! Systematic errors due to model and forward operator cannot be assumed to be smaller than instrument/calibration errors. What you can assume is that the different error contributions have different spatial and temporal characteristics (e.g. the jump in Fig. 2 is very likely related to the calibration changes), which can be used to identify likely error sources.

**Our response:**

We agree with you on this comment. Currently, it is difficult to illuminate the representativeness errors which are related to the NWP models and forward operators. One example of the representativeness error in B was caused by the annular solar eclipse on 21 June 2020 (Fig. S4(b)). The forward operator neglected the abrupt decrease of incoming solar irradiance, leading to underestimated reflectance in the synthetic image. Therefore, the sentence was replaced by "One assumption of the inter-comparison method is that the spatiotemporal characteristics of different error contributions differ so that the O-B analysis can be used to identify different error sources" in the revised manuscript. (L81-82)

**Specific Comment:**

- l. 61: "NWP model errors could be alleviated [] by temporally averaging several instants over a long period of time". Averaging over a long period can help to detect systematic errors, but it cannot "alleviate" them. No DA system works with time-averaged states...

**Our response:**

Thanks for pointing this out. This sentence was deleted in the revised manuscript. Averaging several instants over a long period of time is helpful to detect some systematic errors of the NWP models. Therefore, the spatial distribution of one-month mean precipitation and O-

B biases of reflectance was analyzed. The analysis revealed systematic errors of the CMA-MESO in overestimating the precipitation areas (Fig. 3, L209). In addition, analysis of the one-month mean O-B differences revealed systematic biases of CMA-MESO+RTTOV-DOM simulations over complex terrain areas (Fig. 5, L255).

**Specific Comment:**

- 2.1: The parameterization for unresolved sub-grid clouds was found to be very important in Geiss et al. (2021). How is CMA-MESO handling this and are the subgrid contributions included in the input data for RTTOV?

**Our response:**

Previous studies suggested that the parameterization for unresolved sub-grid clouds was critical to the simulated reflectance (Scheck et al., 2018; Geiss et al., 2021). In this study, the sub-grid clouds were approximated by the meso-SAS shallow-convective cumulus parameterization. The tendency equations of the grid-box mean moist static energy, water vapour mixing ratio, and vertical velocity were related to the transfer equations of related variables at sub-grid scale. The mixing ratio of cloud hydrometeors at sub-grid scale was generated by convective condensation with interactions to gird-scale processes considered. The spatial coverage of the sub-grid clouds within a grid box was depicted by cloud cover, which was diagnosed from the grid-scale humidity following Xu and Randall (1996). The cloud cover derived from the CMA-MESO forecast was included in the RTTOV input to account for the sub-grid contributions and the radiative transfer was solved by using the maximum random overlap method. (L117-125)

**Specific Comment:**

- l. 95/Fig. 5: For the Baum ice clouds you need also the effective ice particle radius -- which parameterization did you use? Do you have an idea why the effect visible in Fig. 5 is so much stronger than what Geiss et al. found for modified ice optical properties?

**Our response:**

The liquid and ice cloud optical properties in RTTOV were parameterized by the "Deff" scheme (Mayer and Kylling, 2005) and the Baran et al. (2014) scheme, respectively. The

effective radius of liquid water clouds ($Re_{liq}$) was calculated following Thompson et al. (2004) and Yao et al. (2018). The effective radius if ice clouds ($Re_{ice}$) was not calculated explicitly since the ice scheme developed by Baran et al. (2014) does not have an dependence on $Re_{ice}$. (L155-159)

In our manuscript, the reference run was configured with Baran scheme, while the experiment run was configured with the Baum scheme which includes a general habit mixture (GHM) of ice crystals. In Geiss et al. (2021), the reference run was configured with the GHM model developed by Baum et al. (2014), whereas the experiment run was configured with solid column scheme based on ice optical properties of Yang et al. (2005). The sensitivity study in our manuscript indicated that the impacts were especially apparent for optically thin clouds (reflectance < 0.2) (Fig. 8(b)) and extended to optically thick clouds. However, Geiss et al. (2021) suggested that changing the ice scheme from GHM to the solid-column scheme only affected the high-reflectance end of the PDF. We did not conduct an inter-comparison study of ice cloud schemes between the solid columns and GHM. But Baum et al. (2014) compared the ice cloud optical thickness retrieved based on the GHM and solid columns and indicated good consistency between two ice models due to their similar asymmetry parameters. However, there should exhibit distinct differences of the optical properties between the Baran scheme and Baum scheme. For example, the Baum scheme was developed based on nine basic ice habits whereas the Baran scheme involves only six ice habits. In addition, the PDFs and the mixing ratio of each habit are different between the two ice schemes. Therefore, the distinct differences between the Baran and Baum schemes should be the main cause to the larger differences than Geiss et al. (2021) between the reference run and experiment run. (L361-380)

**Specific Comment:**

- l. 98/117: How well does "cloudy" in the CLM product correspond to CWP > 0.01kg/m2? This probably rather unclear (no pun intended). The CLM algorithm may even use infrared channels and could therefore "see" clouds that are actually transparent in the visible range (I have not checked that). A more reliable way to define cloudy pixels would be to use reflectance > threshold value (e.g. r>0.2 as in Geiss et al. 2021) or even better reflectance > clear sky

reflectance + threshold value. Exactly the same criterion could be applied to the observed and the synthetic reflectances.

**Our response:**

Thank you for the constructive suggestion to build an equivalent criterion of cloud mask for the simulations and observations. In the revised manuscript, cloud mask was determined by comparing the simulated and observed reflectance with the reflectance simulated by ignoring cloud impacts. (L278-303)

For the synthetic visible image, a pixel was designated to be cloudy if the simulated reflectance $r_{sim}$ satisfies Equation (4). Otherwise, the pixel would be classified to be cloud-free.

$$r_{sim} > r_{sim,clear} \tag{4}$$

where $r_{sim,clear}$ denotes the simulated reflectance when cloud contributions were ignored.

The aerosol contributions were neglected by the simulations since the CMA-MESO forecasts do not provide aerosol information explicitly, whereas the observed reflectance inevitably includes aerosol contributions. Considering the aerosol contributions to the reflectance, a pixel is assumed to be cloudy if the observed reflectance $r_{obs}$ satisfies Equation (5),

$$r_{obs} > r_{sim,clear} + r_{aer}^{75} \tag{5}$$

where $r_{aer}^{75}$ denotes the aerosol contributions to the reflectance of cloud-free pixels, which was set to the upper quartile of $r_{obs,clear} - r_{sim,clear}$ for the preliminarily estimated cloud-free pixels. $r_{obs,clear}$ denotes the observed reflectance for cloud-free pixels, which were preliminarily determined by the FY-4A CLM product. The second-step estimate of cloud-free pixels was determined Equation (6),

$$r_{obs} < r_{sim,clear} + r_{aer}^{25} \tag{6}$$

where $r_{aer}^{25}$ denotes the aerosol contributions to the cloud-free reflectance. Similarly, $r_{aer}^{25}$ was set to the lower quartile of $r_{obs,clear} - r_{sim,clear}$ for the preliminarily estimated cloud-free pixels. The two-step estimate of cloud mask for observed images was performed to maintain equivalent criterion of the cloud mask for synthetic images. It is noted that the first-step estimate of cloud mask should have different representativeness compared with the cloud mask diagnosed from Equation (4). For example, the CLM cloud mask was generated by including

extra infrared observations (Wang et al., 2019) that are much more sensitive to optically thin cloud, which may appear to be transparent in the visible band. Nevertheless, the quartile estimation should mitigate the impacts. On one hand, thin clouds which are transparent in the visible channel whereas are opaque in the infrared channels should contribute insignificant part to $r_{obs}$. On the other hand, the quartile estimation in Equations (4) and (5) discarded 25% samples in estimating the lower and upper quartiles of $r_{obs,clear} - r_{sim,clear}$.

**Specific Comment:**

- End of 2.1: Maybe here would be a good place to add some more information on RTTOV. In contrast to Anonymous Reviewer #1 I do not think RTTOV in the visible range "is still questionable". No forward operator is perfect, but there are several studies (Geiss et al 2021, Lopez & Matricardi 2022) that demonstrate the capabilities and discuss limitations of RTTOV for visible satellite channels. Moreover, RTTOV is used for the operational assimilation of the visible Meteosat SEVIRI channel at DWD, which is a clear indication that RTTOV produces reasonable results. While most of them are based on the MFASIS solver (Scheck et al. 2016, 2022), the latter is just an emulator for the DOM solver used in this study, so error estimates derived for MFASIS present upper bounds for DOM errors. (I guess MFASIS coefficients are not yet available for FY-4A).

**Our response:**

Thank you for this suggestion. We added some comments on this topic in the introduction. (L64-80)

"The inter-comparison method was also applied to satellite visible channels (Geiss et al., 2021; Lopez and Matricardi, 2022; Lopez et al., 2022) to better understand the observation errors and representativeness errors and to provide guidance for the improvements of NWP models and forward operators. Most of the studies performed the radiative transfer simulations based on a software package termed the Radiative Transfer for the Television infrared observation satellite Operational Vertical Sounder (TOVS) (RTTOV) (Saunders et al., 2018). To save computational cost, a method for fast satellite image synthesis (MFASIS) was developed based on a lookup table (LUT) computed with one-dimensional (1D) solver of

RTTOV in rotated Cartesian coordinates to account for some three-dimensional (3D) radiative effects (Scheck et al., 2016; Scheck et al., 2018). To better simulate the tangent linear and adjoint models, a neural network-based forward operator was also developed based on RTTOV simulations (Scheck et al., 2021). Intercomparison of satellite visible reflectance and the equivalents derived from NWP models and MFASIS indicated generally good agreement, and the Bidirectional Reflectance Distribution Function (BRDF) of land surface derived from a monthly mean atlas generated reasonable results in cloud-free conditions (Lopez and Matricardi, 2022). However, neglecting aerosol contributions in the radiative transfer simulations would lead to systematic biases both in cloudy and cloud-free conditions (Geiss et al., 2021). Data assimilation of satellite visible reflectance data based on the MFASIS suggested positive impacts in real-world cases (Scheck et al., 2020). Since March 2023, satellite visible reflectance data have been operationally assimilated in German Weather Service by using the MFASIS forward operator. Existing studies imply the promising expectation that RTTOV could generate reliable visible images if the NWP models were well tuned and the model configurations were optimized."

**Specific Comment:**

- l. 109: It would be good to provide the wavelength

**Our response:**

Done. (L93)

**Specific Comment:**

- l. 116: If you produce output with CMA-MESO every 15 minutes and FY-4A/AGRI starts scanning from the north of the disk to the south at the same times (I am not sure about the scanning strategy), the time difference is probably smaller than 7.5 minutes. Please check...

**Our response:**

The full-disk scanning cycle of AGRI is 15 minutes and the scanning usually starts at 00:00 UTC. In addition, the CMA-MESO forecasts were produced at hourly intervals (e.g., 04:00, 05:00, 06:00, …). Therefore, the maximum allowable time differences between the FY-

4A observations and CMA-MESO forecasts are within 15 minutes to ensure the temporal match. (L166-170)

**Specific Comment:**

  - Fig 2: Is all = cloudy + cloud-free? Then I do not understand how the bias for all can be larger than for cloudy. Or is the third category "uncertain" missing in the plot? Then it would be good to add it. Moreover, the number of pixels in cloudy/cloud-free/uncertain as a function of time would be interesting. Maybe this is helpful for the interpretation: If the jump on 9th of September is related to a changed calibration factor (the constant used to convert the number of detected photons to a radiance) then the change in reflectance bias in cloudy/cloud-free should be proportional to the mean reflectance in cloudy/cloud-free. The fact that the cloud-free bias increases indicates that either the calibration has in fact become worse on September 9th or that before a calibration-related bias compensated a clear-sky bias (e.g. due to an albedo bias).

**Our response:**

  (1) In the original manuscript, all = cloudy + cloud-free + uncertain. In Fig. 7 of the revised manuscript, all = cloudy + cloud-free to avoid misleading. Fig. 7 includes three subpanels. The first panel showed the biases for all pixels (cloudy + cloud-free for both land and sea surfaces) and the corresponding number of pixels. The second panel showed the biases and the number of pixels for land surface, where results for cloudy and cloud-free pixels were presented separately. The third panel showed the biases and the number of pixels for sea surface, where results for cloudy and cloud-free pixels were presented separately. (L331-334)

  (2) The abrupt change of the bias from September 8th to 9th was caused by the measurement calibration processes, which were confirmed by two facts. First, the O-B biases were positively related to the observed reflectance which is proportional to the calibration coefficient. Therefore, an abrupt of the domain-averaged observed reflectance was also revealed. Second, the calibration correction coefficients of FY-4A/AGRI channel 2 were updated by the National Satellite Meteorological Center (NSMC) of CMA at 02:00 UTC on 9 September 2020 (http://www.nsmc.org.cn/nsmc/cn/news/103609.html) (remember that both the observations and simulations were deployed at 06:00 UTC). (L316-321)

(3) Cloud-free biases were reduced after the calibration correction coefficients were updated (Fig. 7(b)), which confirms the effectiveness of the calibration processes. (L321-322)

**Specific Comment:**

- l. 145: If I understand this correctly, your "ensemble" is just seven deterministic model states for different lead times. This is not what anybody in data assimilation would call an ensemble and you are definitely not evaluating an "ensemble forecast" (l. 146). This is really misleading and I also do not see why this "ensemble" should be interesting. If you compute the average of synthetic images for different lead times you will of course get a blurred version of the 3h lead-time image and increase the cloud cover. But no operational data assimilation system I know uses time-averaged states. Is the point here that in a real ensemble DA system you would use the "real ensemble" for the bias correction but as you have none available here you are using the set of deterministic model states, because it has similar properties (the clouds are not exactly at the same locations in all of the members)?

**Our response:**

It is noted that the ensemble forecast here could not represent a real ensemble in any operational ensemble DA systems. On one hand, the number of ensemble members is too small to fully represent the uncertainties of atmosphere states. On the other hand, a more commonly used way to generate an ensemble forecast is to add perturbations to the ICs and LBCs or to combine several forecasts with different combination of microphysical schemes (Li et al., 2015). The simplified ensemble forecast in this study was used mainly because none of a well-tuned ensemble forecast is currently available for the selected area. Nevertheless, synthetic visible images derived from the ensemble forecast should be accompanied with increased cloud cover since clouds are not exactly overlapped for different ensemble members. As a result, the number of matched pixels which are cloudy both for the observations and simulations would be increased, which benefited the bias correction in cloudy regions (see Section 5 for more details). In a real ensemble DA system, a real ensemble would be adopted for the bias correction. (L139-148)

**Specific Comment:**

- l. 147 / 211: Using the overbar for both ensemble mean and spatial average is confusing.

**Our response:**

In the revised manuscript, the overbar specifically denotes the domain average. Synthetic image generated by an ensemble forecast was explicitly noted in the text. (L431, L473-484)

**Specific Comment:**

- Table 1 (deterministic forecast): I am missing a better discussion of the results. The cloudy biases are reduced by 40-60%, but in all cases the bias correction actually *increases* the bias for the clear pixels. For SEP 15 & 17 the bias correction changes the sign of the bias for the clear pixels to negative. As the bias is positive for the cloudy pixel, this leads to compensating biases for the "all" category. Why is the bias correction ineffective for the clear pixels?

**Our response:**

The bias correction method used the systematic biases derived from the cloud-free (or cloudy) pixels for both O and B to estimate error characteristics of cloud-free (or cloudy) pixels only for O. Apparently, the cloud-free (or cloudy) pixels both for O and B are only a subset of those only for O. Therefore, the performance of the bias correction is determined by the representative of the subset of cloud-free (or cloudy) pixels to the corresponding cloud-free (or cloudy) pixels only in the observed images. (L437-442)

The bias correction was tested by two selected cases on September 15th and 17th, 2020. The case on September 1 was deleted due to the errors in observations before September 8th. For the ensemble forecast, the synthetic image was generated by averaging the seven visible images simulated from seven ensemble members. Cloud mask was determined by Equation (4) except that $r_{sim}$ and $r_{sim,clear}$ denotes the reflectance of the ensemble mean synthetic image. For the bias correction based on deterministic forecasts, O-B biases were reduced in most cases, but increased biased were also revealed on September 17th for cloudy regions over sea (Table 1). In contrast, the bias reduction was especially effective when B was derived from ensemble forecasts (Table 2). Since the synthetic image for an ensemble forecast would increase cloud

cover compared with a deterministic forecast, the number of the matched cloudy pixels was increased for an ensemble forecast. As a results, γ derived from ensemble forecasts should represent cloudy bias characteristics better than a deterministic forecast and vice versa, which explains why the biases were increased in some cases based on deterministic forecasts. In cloud-free regions, the original O-B biases were trivial, and the bias correction in cloud-free regions reduced the O-B biased to almost zero. (L443-458)

**Specific Comment:**

- l. 253: What does "cloudy" mean for the ensemble? That in the ensemble mean state CWP>0.01kg/m2?  So potentially clear member pixels contribute to bar{B_cld}?

**Our response:**

For the ensemble forecast, the synthetic image was generated by averaging the seven visible images simulated from seven ensemble members. Cloud mask was determined by Equation (4) (L284) except that $r_{sim}$ and $r_{sim,clear}$ denotes the simulated reflectance from the ensemble mean synthetic image. (L444-446)

**Specific Comment:**

- l. 304ff: There are two YZs -- maybe use the full names.

**Our response:**

The first YZ is Yongbo Zhou, and the second YZ isYuefei Zeng. We used the full names in the author contribution. (L536-541)

**References:**

- Geiss et al. (2021): "Understanding the model representation of clouds based on visible and infrared satellite observations", ACP, Volume 21, issue 16, https://doi.org/10.5194/acp-21-12273-2021

- Lopez & Matricardi (2022): "Validation of IFS+RTTOV/MFASIS0.64-μm reflectances against observations from GOES-16, GOES-17, MSG-4 and Himawari-8", ECMWF Technical

memorandum, DOI 10.21957/l4u0f56lm, https://www.ecmwf.int/en/elibrary/81322-validation-ifsrttovmfasis064-mm-reflectances-against-observations-goes-16-goes-17

- Scheck et al. (2016): "A fast radiative transfer method for the simulation of visible satellite imagery", Journal of Quantitative Spectroscopy and Radiative Transfer, Volume 175, 54-67, https://doi.org/10.1016/j.jqsrt.2016.02.008.

- Scheck, L., Weissmann, M., and Bernhard, M. (2018): "Efficient Methods to Account for Cloud-Top Inclination and Cloud Overlap in Synthetic Visible Satellite Images", J. Atmos. Ocean. Tech., 35, 665-685, doi: 10.1175/JTECH-D-17-0057.1

- Scheck (2020): "A neural network based forward operator for visible satellite images and its adjoint", Journal of Quantitative Spectroscopy and Radiative Transfer, Volume 274, November 2021, 107841, https://doi.org/10.1016/j.jqsrt.2021.107841

**Our response:**

All the papers you recommended were cited in the revised manuscript.

---

## Author Response (AR2)

**Response to reviewers**

**Comment 1:**

- l. 36: "lunched" should be "launched"

**Our response to comment 1:**

Corrected. (**L36, L46**)

**Comment 2:**

- Fig. 6: Which images are model equivalents and which ones are observations? This information is missing.

**Our response to comment 2:**

The figure caption was polished in the revised manuscript. Explanation to each subplot was added. The revised figure caption is "***Figure 6: Synthetic (the first column) and observed (the second column) visible images and the corresponding probability density distribution functions (the third column) for two selected cases. The first panel (a1-a3) is the results for the case at 06:00 UTC on 1 September 2020. The second panel (b1-b3) is the results for the case at 06:00 UTC on 15 September 2020.***". The revised figure caption should avoid the ambiguity. (**L256-260**)

**Comment 3:**

- l. 158 / Fig. 8: You say that you did not need the ice particle effective radius, as it is not required for the Baran scheme. However, you compared the Baran scheme to the Baum scheme, which does require an effective radius. So which effective ice particle radius did you use to generate Fig. 8?

**Our response to comment 3:**

Thank you for pointing this out. The Baran scheme does not have a dependence on the effective radius of ice clouds ($Re_{ice}$). However, the Baum scheme does depend on $Re_{ice}$. Since the state variables of the CMA-MESO model does not include $Re_{ice}$, $Re_{ice}$ was explicitly calculated following Hong et al. (2004) and Yao et al. (2018) to facilitate the radiative transfer simulations based on the Baum scheme. The method was summarized in the following. (**L368-375**)

$$Re_{ice} = \min(11.9 \times 0.75 \times 0.163 \times M_i^{1/2}, 500 \times 10^{-6}) \tag{R1}$$

where $M_i$ denotes the ice crystal mass, which is calculated by Equation (R2),

$$M_i = \frac{\rho_a q_i}{N_i} \tag{R2}$$

where $\rho_a$ denotes the density of air. $q_i$ denotes the mixing ratio of ice crystals. $N_i$ denotes the concentration of ice crystals which was approximated by Equation (R3),

$$N_i = \min(\max(5.38 \times 10^7 (\rho_a \times max(q_i, 10^{-15}))^{0.75}, 10^3), 10^6) \tag{R3}$$

**Comment 4:**

- l. 271: Typical snow albedo values are higher in the visible spectrum (see e.g. Gardner, A. S., and M. J. Sharp (2010), A review of snow and ice albedo and the development of a new physically based broadband albedo parameterization, J. Geophys. Res., 115, F01009). However, taking into account the the snow may not fill the full pixel, you could claim that 0.2 is used as a lower limit for the surface albedo of snow-covered surfaces.

**Our response to comment 4:**

Thank you for this comment and for recommending this paper, which is really helpful. The surface albedo of snow in the visible spectral band varies with the physical properties of snow. With the increase of the average radius of ice grains, the surface albedo is decreased (Gardner and Sharp, 2010). In addition, the surface albedo of dirty snow which includes absorbing particles and old snow which includes some melting water is less than that of pure snow (Xu and Tian, 2000; Gardner and Sharp, 2010). In general, the lower limit for the surface albedo of snow-covered surfaces in China is suggested to be 0.2 (e.g., Fig. 3 of Xu and Tian, 2000, shown below).

[Figure]

**Fig. 3 in Xu and Tian. The spectral distribution of the albedo for different snow state.**

Therefore, the threshold test was performed by the following formula,

$$\omega \leq 0.2 / 3.14 \tag{R4}$$

where $0.2 / 3.14$ denotes the BRDF for a Lambertian radiator. (**L267-274**)

**Comment 5:**

- Section 3.3: It seems to me that to detect instrument problems actually the cloud-free pixels (blue lines in Fig. 7) would be enough -- they clearly show the change in calibration. Only clear sky reflectances are used for these blue lines, so no model state is involved -- right? Is there an advantage of using also the model state (looking at the red/black line in Fig. 7)?

**Our response to comment 5:**

It is true that the time series of the O-B biases for cloud-free pixels could reflect the abrupt change in calibration. This is reflected by the blue lines in Fig. 7. To simulate the TOA reflectance for cloud-free pixels, some of the model state variables derived from the CMA-MESO model were provided to the RTTOV-DOM radiative transfer model. These model state variables include the temperature profile, the humidity profile, the surface pressure, the 2-m height temperature and humidity, the 10-m height U- and V-wind components, and the surface temperature.

The abrupt change was also revealed by the red and black lines in Fig. 7. Since the O-B biases were positively related to the observed reflectance which is proportional to the calibration coefficient, the mutant signal was amplified when cloudy pixels were involved (remember that the clouy reflectance is generally larger than cloud-free reflectance). Therefore, involving cloudy pixels for monitoring the performance of the instrument should have some advantages. When cloudy pixels were involved, the state variables provided to RTTOV-DOM not only include the clear sky variables mentioned above, but also include cloud variables, i.e., the mixing ratio of cloud droplets, the mixing ratio of rain, the mixing ratio of ice, the mixing ratio of snow, the mixing ratio of graupel, the grid-scale cloud fraction, and the effective radius of liquid water clouds explicitly calculated based on some parametrizations. (**L324-325**)

**Comment 6:**

- Table 2, caption: "Land+Scean" should be "Land+Sea"

**Our response to comment 6:**

Corrected. (**Table2 in Page 20**)

**Comment 7:**

- Fig. 10: A background grid or zero lines would be very helpful here to better see the asymmetry.

**Our response to comment 7:**

We added background grid to Fig. 10. (**L528-531**)

**Comment 8:**

- l. 435ff: So if I understand correctly, there are four bias correction coefficients, gamma_land,cloudy, gamm_land,clear, gamma_ocean,cloudy and gamma_ocean,clear. It would be useful to state that explicitely here.

**Our response to comment 8:**

Yes you are right. There are four bias correction coefficients corresponding to two underlying surfaces (land and sea) and two cloud masks (cloud and cloud-free). In addition, some of the cloud masks derived from the FY-4A CLM product could be also "uncertain". Therefore, two extra bias correction coefficients were involved to deal with the "uncertain" scenarios for land and sea separately. Therefore, the bias-corrected reflectance $O^{'}$ is calculated by Equation (R5),

$$O^{'} \ = \ O(1 \ + \ \gamma_{clm}^{sfc}) \tag{R5}$$

where $\gamma_{clm}^{sfc}$ denotes the bias correction coefficient. The subscript $clm$ denotes cloud mask, which is either cloud-free ($clr$), or cloudy ($cld$), or uncertain ($uct$). The superscript $sfc$ denotes the surface type that is either land or sea. To be specific, $\gamma_{clm}^{sfc}$ represents one of the six bias correction coefficients including $\gamma_{clr}^{land}$, $\gamma_{cld}^{land}$, $\gamma_{uct}^{land}$, $\gamma_{clr}^{sea}$, $\gamma_{cld}^{sea}$, and $\gamma_{uct}^{sea}$. (**L448-462**)

**Comment 9:**

- l. 470ff: The bias correction coefficients are now called mu instead of gamma. It would be good to stick to one name.

**Our response to comment 9:**

Thank you for pointing this out. In the revised manuscript, $\mu$ was replaced by $\gamma$ here and elsewhere. (**L448-462; L509-527**).

In addition, the simulated reflectance, originally denoted by *r*, was denoted by "*B*" in the revised manuscript. The modification was to maintain consistency with the so-called "*O-B*" analyses in section 5.

**Comment 10:**

- l. 443: "For the ensemble forecast, the synthetic image was generated by averaging the seven visible images simulated from seven ensemble members.". I do not understand. I thought you would just use the additional enemble members to get more matches (pixels that are cloudy in both O and B or cloud-free in both O and B) that can be used for the computation of mu_clr, mu_cld (l. 472/473). However, I cannot see how you would use the ensemble average for this purpose. As an example, suppose the model is actually unbiased and produces clouds with the same reflectance distribution as the

observation, and the clouds are just at the wrong locations. Then in the ensemble average the reflectances will be lower (because if there is a cloud, it will in general not be present in all members) and if you use these low, ensemble averaged reflectances in the computation of mu_cld you will get a positive, potentially even large value (and not the value 0 that would be correct for no bias). Are you really using the ensemble average in this way and if yes, why is it working? Or did I misunderstand how you use the ensemble average? Then please provide an equation with the ensemble average reflectances that clearly shows what you are doing.

**Our response to comment 10:**

For the Ensemble Kalmam Filter (EnKF) method which has been widely used for the DA of satellite radiance data, the observation increments were calculated using the ensemble mean in the observation space (e.g. Equation (5) of Zhou et al., 2022).

$$\Delta y_n = \left(y_n^{\mathrm{p}} - \bar{y}_{\mathrm{p}}\right)\left(\sigma_{\mathrm{u}}/\sigma_{\mathrm{p}}\right) + \bar{y}_{\mathrm{u}} - y_n^{\mathrm{p}}, \; n = 1, \dots, N \tag{R6}$$

where $\Delta y_n$ denotes the observation increment for the $n$th ensemble member, $y_n^{\mathrm{p}}$ the first guess of the observed variable for the $n$th ensemble, $\bar{y}_{\mathrm{p}}$ the ensemble mean of the first guess of the observed variable, $\sigma_{\mathrm{p}}$ the first-guess sample error standard deviation of the observed variable, $\sigma_{\mathrm{u}}$ the updated standard deviation of $\sigma_{\mathrm{p}}$, $\bar{y}_{\mathrm{u}}$ the ensemble mean of the posterior estimate (i.e., the analysis) of the observed variable.

Therefore, to maintain consistency with the ensemble-based DA methods, the bias correction method should be performed based on the ensemble mean of the first-guess reflectance, denoted by $\overline{B_{clm}^{sfc}}$, which was generated by Equation (R7),

$$\overline{B_{clm}^{sfc}} = \frac{1}{N_{ens}} \sum_{l=1}^{N_{ens}} B_{clm}^{sfc}(l) \tag{R7}$$

where $N_{ens}$ denotes the number of ensemble members, i.e., seven in this study. $l$ is the index for an arbitrary ensemble member. (**L463-478**)

We agree with your comments that the ensemble averaging could decrease the reflectance for a pixel classified to be cloudy for a deterministic forecast in cases where the cloud did not occur for all ensemble members due to the displacement errors. As a result, the bias correction coefficient estimated by the ensemble forecasts is larger than that estimated by a deterministic forecast due to the underestimated first-guess reflectance (Table 1 and Table 2). In other cases where clouds occur for all the ensemble members, the uncertainty of the ensemble mean should be smaller than the uncertainty of a single ensemble member. In general, the number of the matched cloudy pixels was larger for the ensemble forecast than the deterministic forecast. Since the O-B departure was dominated by the cloudy error

characteristics, the PDF of the bias-corrected O-B departure satisfied the unbiased Gaussian function better for the ensemble forecast than the deterministic forecast. (**L479-486, L500-596, Fig. 10, L528**)

**Comment 11:**

- Section 5: To demonstrate that you need to distinguish between land/ocean and cloudy/cloud-free, it would be great to provide values for the bias correction coefficients for the two dates. And to show how helpful the ensemble is, it would be good to provide information on the the number of matching pixels with / without ensemble.

**Our response to comment 11:**

We added the bias correction coefficient $\gamma$, the number of matching pixels $N_{match}$, and the number of pixels in the observed images $N_{obs}$ for cloudy and cloud-free scenarios over land and sea surfaces. (**Table 1 in Page 19 and Table 2 in Page 20**)

**Other changes**

In addition to the changes mentioned above, we corrected some typo errors and polished the writings throughout the manuscript to avoid ambiguities and misunderstandings. All these revisions were marked in red in the track-changes file.